

# Influence of atmospheric waves and deep convection on water vapour in the equatorial lower stratosphere seen from long-duration balloon measurements

Sullivan Carbone[1], Emmanuel D. Riviere[1], Mélanie Ghysels[1], Jérémie Burgalat[1], Georges Durry[1], Nadir Amarouche[2], Aurélien Podglajen[3], Albert Hertzog[3]

[1] Groupe de Spectrométrie Moléculaire et Atmosphérique (GSMA, CNRS UMR 7331), Université de Reims, UFR Sciences Exactes et Naturelles, Moulin de la Housse B.P. 1039, 51687 Reims Cedex 2, FRANCE
[2] INSU Division Technique, 1 place Aristide Briand, 92195 Meudon cedex, FRANCE
[3] Laboratoire de Météorologie Dynamique (LMD/IPSL), Sorbonne Université, École polytechnique, Institut polytechnique de Paris, École normale supérieure, PSL Research University, CNRS, Paris, France

*Correspondence to*: sullivan.carbone@univ-reims.fr

**Abstract.** The STRATEOLE 2 project consists of 3 campaigns of stratospheric superpressure balloons released from the Seychelles and intended to fly over the equatorial belt transported by winds during 3 to 4 months. During the two campaigns which have already been carried out, (2019/2020 and 2021/2022) five Pico-STRAT Bi Gaz spectrometers have been released in order to measure *in situ* water vapour, methane ($CH_4$) and carbon dioxide ($CO_2$) around 18.5 km and 20.5 km. In this paper, we have developed a methodology based on the calculation of *in situ* water vapour and temperature anomalies to estimate the modulation of water vapour due to atmospheric waves and deep convection. The calculation of Pearson correlation coefficient is performed between averaged ERA5 reanalysis temperatures and in situ water vapour anomalies. In case of a monotonic vertical gradient of water vapour, the absolute value of the Pearson's r is high (typically 0.65) when atmospheric waves are a predominant factor of modulation. This is the case for the flight C0_05_TTL2. In case of other flights, we notice a decrease of the Pearson's r absolute value which can be explained by the change in time of the vertical gradient of water vapour, and large convective systems with turrets overshooting the tropopause. This is the case for the flight C1_15_TTL4 which flew over the Rai typhoon (Pearson's r of 0,31 due to both contributions)

## 1. Introduction

Water vapour is the most important greenhouse gas on Earth. In the stratosphere, water vapour plays a major role in the chemical equilibrium, especially in the ozone ($O_3$) budget, where it is the main source of Hydroxyl radical (OH). Furthermore, it plays a significant role in the global radiative budget, especially in the light of an increase of water vapour during most of the past decades (Solomon et al., 2010; Dessler et al.,2016). Stratospheric water vapour has increased in the middle stratosphere at a rate of 0.5- 1%/year  (Oltmans et al., 2000; Rosenlof et al., 2001; Scherer et al., 2008; Hurst et al., 2011) whereas a trend is difficult to estimate nearby the tropopause due to the variability of its height and the influence of dynamic processes which



modulates the abundance of water vapour. Observational studies have shown that an increase of stratospheric water vapour could lead to a warming of the mean surface temperature (Forster and Shine, 1999; Solomon et al., 2010; Wang et al., 2017). Even a small change of water vapour (less than 1 ppmv) in the lower stratosphere represent an important source of the decadal variability in the surface temperature. Aside from transport and modulation processes, stratospheric methane oxidation is the major source of water vapour in the stratosphere (Texier et al., 1988). The increase of stratospheric water vapour can partially

be explained by the intensification of methane oxidation and to the global increase of methane injected into the stratosphere. However, the variability of stratospheric water vapour observed during the 90's and early 2000 does not follow the increase of methane during the same period (Rinsland et al., 2009; Dlugokencky et al., 2009; Angelbratt et al., 2011). Numerous uncertainties thus remain in the understanding of physical, dynamical and chemical mechanisms taking place in the stratosphere which drive the stratospheric water vapour abundance.

The slow ascent above the net zero radiative heating level (14-15 km) in tropical tropopause layer (TTL) is a major mechanism of water vapour stratospheric variability. During the ascent, air masses first experience decreasing temperatures until reaching the cold point tropopause. Saturation with respect to ice may be reached, leading to the formation of small ice particles which can sediment and dry the air entering the stratosphere (Gettelman et al., 2000). As a consequence, stratospheric water vapour in the tropics is largely linked to the coldest temperature experienced through the slow ascent (the Lagrangian cold point;

Fueglistaler et al., 2005). The cold point temperature and height exhibit a seasonal variation as a main driver of stratospheric water vapour variability. Deep convection is another important process in the modulation of stratospheric water vapour in the case where it reaches the lower stratosphere (LS) by overshooting convection. Depending on the thermodynamic properties of the LS surrounding the overshoot, these ice particles sublimate and thus hydrate locally the LS (Grosvenor et al., 2007; Chemel et al., 2009; Chaboureau et al., 2007; Behera et al., 2022). Conversely, if the tropical LS is saturated with respect to ice,

particles grow by solid condensation of surrounding water vapour, and sediment if large enough (Hassim and Lane, 2010; Danielsen, 1982). Furthermore, the intensity of deep convection is a determinant factor of the incoming stratospheric water vapour because it modifies the CPT height and, as a consequence, the TTL temperature (Fueglistaler et al., 2009). The impact of the overshooting deep convection on the stratospheric water vapour budget is not well quantified at the global scale due to the difficulty to take into account the variability of the impact of the overshoots at a local scale and to have reliable satellite-

borne statistics of stratospheric overshoots needed to upscale their impact.  Existing climatologies are based on observations from satellite, which may miss the peak time of overshooting activity for the continental tropical deep convection (Iwasaki et al., 2010) or the top altitude of the overshoot (not necessarily reaching the stratosphere: Rysman et al., 2017). On the other hand, the convection permitting simulation of Dauhut and Hohenegger (2022) shows that during the period of 1 August to 9 September 2016, deep convection contributed 11% to the increase in stratospheric water vapour between 10°S-30°N. Another

modelling study suggested that deep convection could contribute in the order of 20 to 50% of the increase of stratospheric water vapour at the end of the 21st century (Dessler et al., 2016).

In tropical regions, atmospheric waves, usually generated by deep convection, also modulate water vapour: wave-induced temperature perturbations may help to reach saturation with respect to ice. Similarly to the slow ascent in the TTL, ice





formation followed by sedimentation can dry the stratosphere. Several modelling studies have investigated the impact of waves

on cirrus formation and stratospheric drying (Jensen and Pfister, 2004; Ueyama et al.,2015; Dinh et al.,2016; Corcos et al., 2023; Schoeberl et al., 2019). Waves also indirectly play a role through the Quasi-Biennial Oscillation (QBO). This oscillation is mainly explained by the interaction between Kelvin waves (and to a lesser extent gravity waves) with the mean flow (Dunkerton, 1997). It was shown that during boreal winter between 2015-2016 a westerly phase of the Quasi-Biennial Oscillation (QBO) and the warm El Niño–Southern Oscillation (ENSO) called El Niño, moistened the lower stratosphere with

positive anomalies of about 20% (Diallo et al., 2018).

The interplay between the above-mentioned processes, occurring at different scale and with importance that may depend on the location, complicates the understanding of the stratospheric water vapour budget. In order to better understand these couplings, the STRATEOLE-2 project consists of in situ observations near the equatorial tropopause with a suite of instruments flown on-board long duration superpressure balloons. In this frame, five Pico-SDLA instruments have been flown (Pico-

STRAT BI Gaz), performing *in situ* measurements of water vapour, methane and carbon dioxide at a temporal resolution of 4 to 12 minutes. The large spatial and temporal extent and high resolution of the measurements open the possibility to link the observed water vapour variability to the influence of atmospheric waves (of different scales) and deep overshooting convection wherever in the equatorial belt. We rely on the calculation of local in situ water vapour anomalies to study the variability of water vapour at different scales. Water vapour anomalies are calculated as the difference between the Pico-STRAT *in situ*

mixing ratio and the mean regional average of satellite borne water vapour fields from the Aura MLS (Micro Limb Sounder) instrument.

The paper is organized as follow: section 2 gives an overview of the Strateole 2 project and of the dynamical context in which the flights took place.  Section 3 presents briefly the Pico-STRAT Bi Gaz (hereafter, Pico-STRAT) and Aura MLS instruments analyzed here. Section 4 describes the methodology employed for the analysis with the sensitivity tests specific to each flights.

Section 5 presents the main results concerning the influence of wave on in situ water vapour. Section 6 deals with the signature of deep convection.

## 2. Balloon and space borne datasets

### 2.1. STRATEOLE 2 long-duration balloon flights

STRATÉOLE 2 is a project funded by the Centre National d'Etudes Spatiales (CNES, France) and the National Science

Foundation (NSF, USA). This project aims at studying dynamical processes in the equatorial lower stratosphere, such as the forcing of the quasi biennial oscillation (QBO) by different types of waves, the formation and life cycle of TTL aerosols and cirrus, as well as the dehydration of the air entering the stratosphere. A final objective is the validation of the AEOLUS satellite wind products at the tropics. STRATEOLE 2 relies on 3 long-duration campaigns where a flotilla of superpressure balloons are launched from the Seychelles Islands.  Such balloons are nearly Lagrangian platforms carrying scientific instrumentation



for flights of several weeks in the TTL and lower the stratosphere between 18 and 20 km of altitude. They evolve on isopycnic surfaces, meaning that the air density remains constant (at first order) during the flight. The two first campaigns took place in 2019/2020 and 2021/2022 respectively. A third one is planned in 2025/2026.

Once at float, the balloons are driven by the winds, and therefore principally drift either eastward or westward, depending on the QBO phase.

The balloons are able to carry up to 15 kg of scientific instrumentation, allowing to probe several meteorological and chemical variables (wind, pressure, temperature, aerosols, clouds, water vapour, other gases etc.) *in situ*. These instruments are powered using batteries located in the Zephyr gondola, which are recharged by solar panels. In the frame of this paper, we only focus on processes playing a role in stratospheric water budget. Five flights of Pico-STRAT have been already released, allowing to measure *in situ* water vapour, $CO_2$ and $CH_4$ mixing ratios. The trajectories of the flights are shown in Figure 1.

One flight (C0_05_TTL2) measuring water vapour and $CO_2$ was launched during the first campaign, circumnavigating at the equator with an average altitude of 19 km. It flew for 79 days and has evolved in the wet phase of the Tape Recorder. During the second campaign, three balloons have flown at an average altitude of 18.5 km measuring water vapour and $CO_2$ (C1_03_TTL4 and C1_15_TTL4) and water vapour and $CH_4$ (C1_07_TTL4). Another instrument has been flown at an average

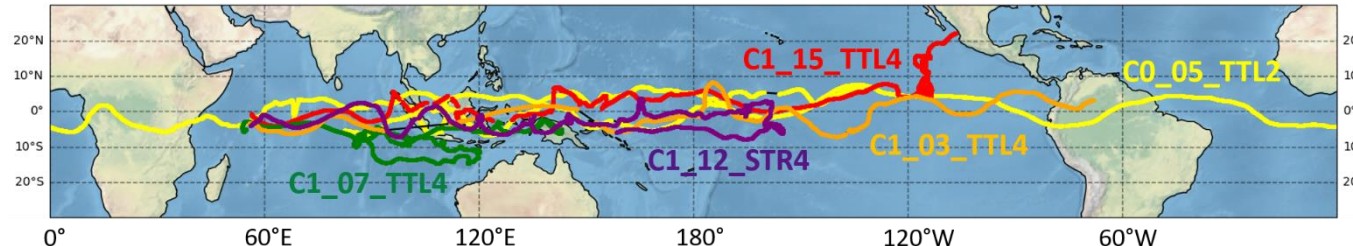

**Figure 1: Balloon trajectories of the flights carrying the Pico-STRAT instrument during the first two campaigns. The balloon trajectory in yellow belongs to the first campaign and the others belong to the second campaign.**

altitude of 20.5 km thus above the TTL top altitude, measuring water vapour and $CO_2$ (C1_12_STR4). They lasted between
31 days and 56 days, essentially overpassing the Indian and the Pacific Oceans. Figure 2(a) shows the MLS v5 water vapour anomaly calculated from the difference between the zonal mean between 10°S and 10°N and five years mean (06/2017 - 07/2022), thereby displaying the tape recorder signal.





**Figure 2: Zonal mean water vapour anomaly of MLS v5 between 10°N and 10°S from July 2018 and July 2022 as a function of air pressure (vertical axis) and time. Panel (a) shows all five flights considered. Panel (b) shows a zoom on the first campaign period with trajectory of the flight C0_05_TTL2 superimposed. Panel (c) shows a zoom on the second campaign period with trajectories of flight C1_03_TTL4, C1_07_TTL4, C1_15_TTL4 and C1_12_STR4 superimposed.**

It shows the alternation of wet and dry phases due to the modulation of the tropopause temperature implying different amounts

of water vapour entering the stratosphere and transported upward by the ascending branch of the Brewer-Dobson circulation.

The trajectories (time/pressure) of the Pico-STRAT balloons are superimposed. Figure 2 (b,c) shows a zoom for the first and



second campaign, respectively. It can be seen that all of the balloons evolved in the wet phase of the tape recorder. However, it can be noticed that depending on the balloon flights, the flight occurred during either the decreasing phase of the wet phase (C0_05_TTL2 and C1_15_TTL4), or during a near steady phase (C1_03_TTL4 and C1_07_TTL4).


## 2.2. Balloon borne Pico-STRAT Bi Gaz

Pico-STRAT Bi Gaz (Ghysels et al, 2024) is an heritage of the former SDLA (Durry and Megie, 1999a), micro-SDLA (Liu et al., 2010) and Pico-SDLA instruments (Ghysels et al., 2011; Ghysels et al., 2012). Pico-STRAT Bi Gaz is a dual-gas tunable diode laser spectrometer, designed to monitor *in situ* water vapour, carbon dioxide and methane in the troposphere and the

stratosphere from large and medium-size open stratospheric balloons, by direct absorption spectroscopy.

The general design of the PicoSTRAT Bi Gaz Bi Gaz is based on the Pico-SDLA instrument (Durry et al., 2008). Water vapour is probed using an antimonide laser diode emitting at 2.63 µm, while carbon dioxide is probed at 2.68 µm and methane at 3.24 µm, regions of strong absorptions within fundamental bands. This allows to dramatically reduce the optical path length and thereby enlighten the instruments. Water vapour is probed over a 1-m path length, $CO_2$ over 50 cm, and methane over

2.5 m in ambient air. The diode laser current is modulated in order to tune the laser frequency, thereby allowing to scan the desired molecular transition. The mixing ratio is extracted from the atmospheric absorption spectrum using a non-linear least-squares fitting algorithm applied to the full line shape, based on the Beer-Lambert law and in conjunction with *in situ* pressure and temperature measurements (Durry and Megie, 1999a). The molecular line shape is modelled using a Voigt profile (VP) in the case of water vapour using line parameters from the HITRAN database. The $CO_2$ and $CH_4$ spectra are modelled using more

advanced profiles, fed with laboratory-based line parameters including temperature dependences (Ghysels et al, 2013a, b). For $CO_2$, a Rautian line profile is used while line mixing is added to a Rautian profile in the case of methane. Retrievals are filtered such as the fitting residuals are consistent with the instrument noise. The *in situ* spectra are taken at 1 s intervals. During that interval, 200 ms are devoted to record the elementary atmospheric spectrum (within this time frame, 5 spectra are recorded), which comprises 256 data points. The remaining 800 ms are used to record the atmospheric pressure and temperature, the GPS

data, and the status of the instrument (internal temperatures, electronics gains, laser current and temperature, etc.).

In order to suit the Stratéole 2 specific operational requirements, the full instrument's electronics, which drives the lasers and acquires the data, is contained in the Zephyr gondola. Doing so allows to keep the electronics at a safe operating temperature without requiring additional power. The optical cell, the laser diodes and the detectors are deported below the Zephir gondola, at a 2-m distance of Zephyr. In this configuration, the electronics module is connected to the lasers and the detectors using 2.5-

m shielded cables. Due to weight limitations, such distance is found to be the best compromise though, impacting daytime, and to a lesser extent, nighttime, measurements especially in the case of water vapour. The analysis of water vapour time series allows to demonstrate that nighttime measurements are free from contamination 5 to 7 days after the launch. In this case, the background mixing ratio during nighttime follows, to some extent, the temporal trend of MLS v5 observations.



### 2.3. Space borne Microwave Limb Sounder (MLS)

MLS is an instrument onboard the Aura satellite measuring upper tropospheric and lower stratospheric constituents from thermal emission at different bands with microwave limb sounding system. Water vapour is measured at 190 GHz on 55 pressure levels between 1000 hPa and 0.001 hPa. The water vapour measurement precision, which drives the measurements dispersion, is 7% at 83 hPa and 6% at 68 hPa (Livesey et al, 2022). In the frame of the present study, we use MLS v5 datasets (Livesey et al., 2020). The main differences between MLS v4 and v5 water vapour products are a reduction of an estimated

20% dry bias below the tropopause (typically 100 hPa), and partial amelioration of a slow positive drift seen in comparisons between MLS and other observations of water vapour in the years since 2010. The extent to which this reduces the drifts reported by Hurst et al., 2016 remains to be investigated.

In the present study, the MLS v5 water vapour measurements are filtered based on the recommendations in Livesey et al., (2020).


### 2.4. Himawari-8 Cloud Top Height products (CTH)

Himawari-8 is the new generation of Japanese geostationary meteorological satellite. It was launched in October 2014 by Japan Meteorological Agency and by Meteorological Satellite Center (JMA/MSC). Its main instrument is an imager named AHI (Advanced Himawari Imager: Bessho et al.,2016) which includes 16 channels (3 visible, 3 near-infrared and 10 infrared) with

respective resolutions of 0.5, 1 and 2 km. Cloud Top Height products are deducted with an algorithm adopted by European Organisation for the Exploitation of Meteorological Satellites Nowcasting Satellite Application Facility (EUMETSAT NWC SAF). This algorithm generates CTH products by using different AHI observations, a radiative transfer model (RTTOV), temperature and humidity vertical profiles obtained with the Numerical Weather Prediction models (NWP) and by using cloud-type data deduced from cloud-type and phase product (Kouki et al., 2016). It is applicable for all imagers on board

meteorological geostationary satellites and uses the lowest resolution of these imagers. In this study, we use CTH products in order to validate the presence of deep convections which could be linked to water vapour anomalies.

## 3. Methodology

### 3.1. Principle

The methodology we have developed relies on the calculation of local anomalies which are obtained as the differences between nighttime i*n situ* water vapour measurements and unbiased MLS v5 values averaged in the same area around the same date. They can be seen as the contrast between the *in situ* measurements and expected climatological mixing ratios around a given location. The time (20 days) and space (typically 400 km) averages are such that the anomalies highlight the area impacted by overshooting deep convection, which occurs on a few km$^2$ and on a 30-minutes timescale, as well as atmospheric waves in the





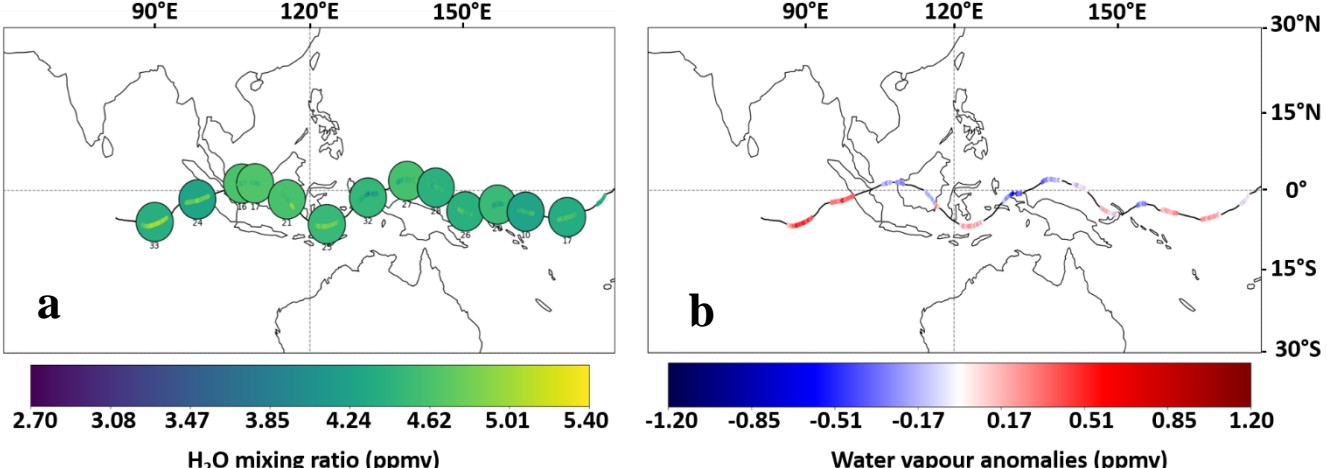

**Figure 3: (a) A part of C0_05_TTL2 night-time trajectory from (12/12/2019) to (12/24/2019) with the** *in situ* **water vapour measurements color-coded. The circles show the location of selected MLS v5 water vapour profiles used to calculate a local mean climatology of water vapour ± 10 days around each night of the flight. The circles are color-filled with the mean MLS v5 water vapour mixing ratio. (b) Corresponding anomalies of water vapour as a result of the difference between** *in situ* **Pico-STRAT measurements and the local mean MLS v5 value in the circle. The thin black line corresponds to the balloon trajectory during daytime.**

water vapour budget. For the rest of the analysis, we consider only the Pico-STRAT nighttime water vapour data. Indeed, daytime measurements suffered from outgassing during a large part of the flights of the (tropospheric) water vapour molecules trapped onto the balloon envelope or the Zephyr gondola walls and released when they are illuminated by the sunlight. We identified night-time measurement using the solar zenith angle (sza) coming from the GNSS receiver onboard. A threshold sza of 95° (typical sunset at 20 km altitude) has been selected for each flight. A scatterplot of water vapour mixing ratios as a

function of sza shows a severe drop in the dispersion of the water mixing ratio above 95° (not shown), indicating that the instrument measures environmental values.

The MLS records used in this study are first selected following space and time collocation criteria. Regarding the space criterion, we have chosen a circle centred on the position of the balloon at the middle of the night. The circle geometry is independent of the direction of the balloon trajectory and is applicable for any kind of trajectory shape. The circle radius is

chosen constant for one given flight and so that it encompasses most of the time the distance browsed by the balloon during a single night. The chosen radius is also a compromise between this distance and the total amount of satellite data within the circle, allowing a sufficient amount of data for averaging. An additional constraint is the insurance that the instrumental bias between Pico-STRAT and MLS v5 remains within the instrumental uncertainties. A discussion around the instrumental bias is given in the following subsection.

The temporal criteria is of major importance since it has the largest impact on the calculation of the anomalies. We have selected the temporal extent so that the impact of large-scale equatorial atmospheric waves is smoothed out. We therefore select 20 days (±10 days around each of the balloon nights) which is longer than the longest wave periods during STRATEOLE 2 as seen from ECMWF ERA5 reanalysis, see also Fathullah et al., 2017. Finally, the MLS mean value within a circle is





interpolated on the mean pressure level of the balloon for a given night. Figure 3 illustrates the results from the selection
process for the flight C0_05_TTL2, from 12 to 24 December 2019 showing mean MLS values and in situ Pico-STRAT
measurements (panel a) and the corresponding water vapour anomalies (panel b). In Figure 3a, the two westernmost circles
show that the *in situ* measurements from Pico-STRAT are wetter than the mean MLS value in the circle, resulting in
corresponding wet anomalies for the two corresponding nights, as shown in Figure 3b.

## 210  3.2. MLS $H_2O$ bias estimation and anomaly calculation

Satellite retrievals are based on mathematical processing and physical interpretations of the observed atmospheric radiances.
Simplifications and small inaccuracies in the design of the instrument and in the algorithms used to process datasets lead to
errors and uncertainties. In the case of MLS v5 $H_2O$ product, the 0.2-1%/year drift identified in v4 products (Hurst et al, 2016)
has been partially corrected and, due to the short-term of the *in situ* observations, this effect is assumed to be small by
comparison with the Pico-STRAT Bi Gaz uncertainty. Additionally to the long-term drift of MLS, bias can be observed with
*in situ* measurements. If the MLS v4 $H_2O$ products have been thoroughly evaluated (Yan et al., 2016; Sheese et al., 2017;
Hurst et al., 2016), the evaluation of the last release v5 needs to be done. In this subsection, we assess the biases between both
Pico-STRAT and MLS v5 $H_2O$ products to build a coherent dataset for our analysis. The full MLS v5 validation study will be
the aim of a forthcoming paper.

The temporal coincidence criterion used for assessing the space-borne water vapour bias is a compromise between a sufficient
closeness in time with the balloon observations and a sufficient amount of MLS data for the statistical analysis. In this line,
the temporal criteria is set to ± 1 day around the balloon mean position overnight. The spatial criteria is the same as used for
the calculation of the anomalies (i.e. 400 km radius around the mean balloon position overnight).

The bias is computed as the difference between third-order polynomials fitted on the MLS and Pico-STRAT raw time series.
The resulting bias is then subtracted from the original MLS values. Figure 4 shows the time series of night-time water vapour
mixing ratios from Pico-STRAT (C0_05_TTL2 flight in 2019/20) and MLS over a period of one day around each night together



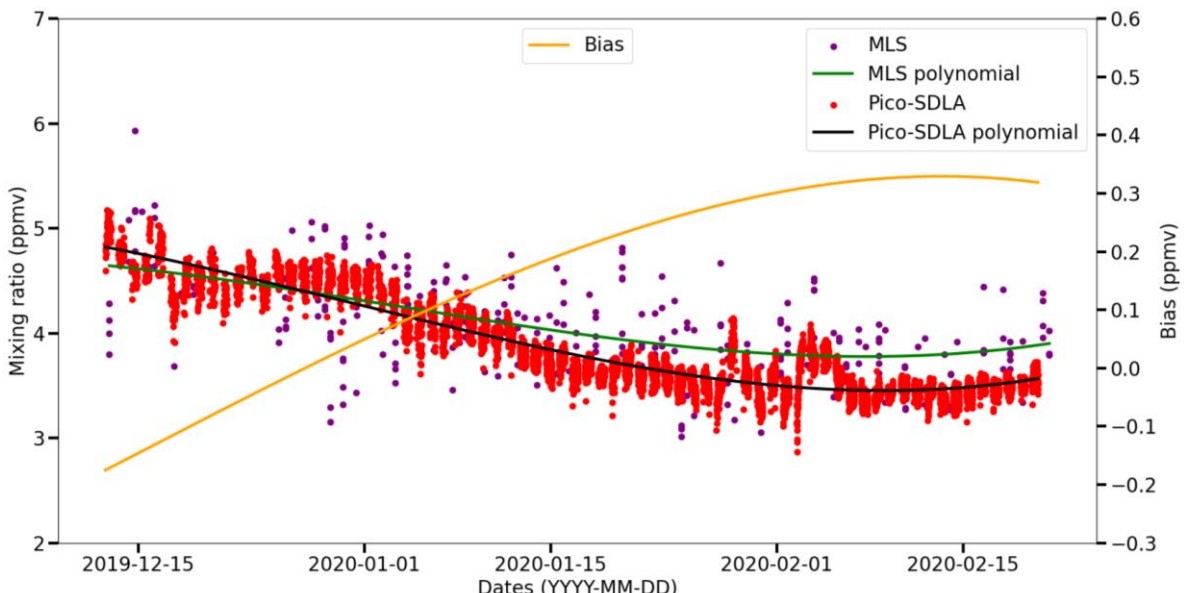

**Figure 4: C0_05_TTL2 flight time series of Pico-STRAT Bi Gaz water vapour mixing ratio (red scatters) and MLS v5 (purple scatters) within circles of 400-km radius and ± 1 day around a given night-time. The associated third-order polynomial of Pico-STRAT Bi Gaz is shown in black line and MLS polynomial in green line. The instrumental bias between both MLS and Pico-STRAT Bi Gaz is shown in yellow line and is calculated from the difference between the two polynomial fits.**

with the corresponding bias. The choice of the space collocation criterion (i.e., the radius circle) has an impact on the representativeness of the calculated bias. In the following, we study the influence of the radius on the anomalies reported. First, sensitivity studies show that for circles with a radius ranging from 400 km to 800 km, the mean bias (calculated over ± 1 day)

remains almost constant, with a variability within 60 ppbv. Therefore, the standard deviation of the water vapour anomalies over one given night should remain almost constant whatever the chosen radius. Figure 5 shows the standard deviation per night for each radius from 400 km to 800 km with a step of 100 km for the C1_07_TTL4 flight. As expected, the standard deviation of the anomalies does not significantly vary for each radius excepted for days 1, 6 and 23. These specific cases can be explained by a local strong variability of MLS water vapour values over ± 2 days. Such cases occur for less than 2% of the

total dataset.



Secondly, the choice of a given radius may change the calculated anomaly, and potentially its sign, since it takes into account the variability of water vapour over ± 10 days, thereby potentially including the contribution of large-scale dynamics. The variability of the anomaly as a function of the radius is given in Table 1 and permits to estimate the uncertainty of the reported

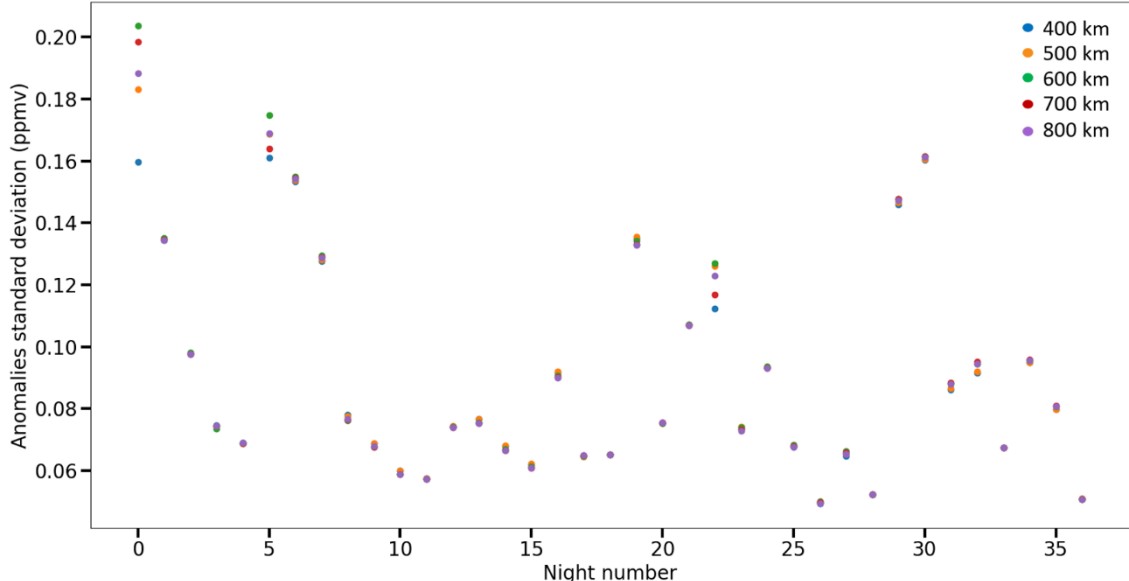

**Figure 5: Standard deviation of water vapour anomalies for each night of the C1_07_TTL4 flight, color-coded as a function of the circle radius used to calculate the anomalies ranging from 400 km to 800 km with a step of 100 km.**

anomalies. The uncertainty of the reported anomalies scales from 0.025 ppmv (for the C0_05_TTL2 flight in 2019-20) up to 0.055 ppmv (for the C1_15_TTL4 flight). First, the amplitude of the uncertainties remains within the Pico-STRAT Bi Gaz and MLS measurement uncertainty, thereby demonstrating the robustness of the anomaly estimation. Secondly, as will be shown in the next sections, these uncertainties in the water vapour anomaly is typically one order of magnitude smaller than the anomalies themselves (several tenths of ppmv). Last, one can notice that the uncertainty is the lowest for the TTL2 flight. Indeed, during this flight, the trajectory of the balloon was strongly zonal, remaining within the ± 8° latitude range over the whole flight. In such case, the latitudinal variability of water vapour at 18.5 km is low, leading to small variability. In the case of most other flights, the latitudinal variability is stronger, so that the MLS variability for circles further from the equator is expected to be stronger.

| Flight | C0_05_TTL2 | C1_03_TTL4 | C1_12_STR4 | C1_07_TTL4 | C1_15_TTL4 |
|---|---|---|---|---|---|
| Standard deviation (ppmv) | 0,025 | 0,041 | 0,041 | 0,035 | 0,056 |

**Table 1: Standard deviation of the mean anomalies for varying colocalization radii for each flight.**



### 3.3. Vertical gradient of water vapour

The vertical gradient of water vapour will be used to discriminate whether air masses are only vertically displaced under the influence of an atmospheric wave or if local variations in water vapour are related to direct injections by deep convection. In our study, the vertical gradient of $H_2O$ has been determined from the calculated averaged water vapour profile from MLS v5. In this subsection, we then compared the *in situ* Pico-STRAT vertical profiles obtained during balloon depressurization events to the mean calculated MLS profiles in order to validate the MLS vertical gradient of water vapour. Depressurizations events occur at night when the balloons fly above cold cloud tops (i.e. deep convection), which radiatively cool the helium inside the balloon, reducing the superpressure and inducing a drop of the balloon altitude (typically of 800 m to 1.5 km) followed by a return to the initial altitude (following sunrise at the latest). These events provide opportunities to measure the vertical profile of water vapour within the altitude range explored at depressurization. Figure 6 (a,b) shows vertical profiles of water vapour from MLS (black curves) ± 2 days around three depressurisation events that took place during the flight C1_07_TTL4(2) and C0_05_TTL2 (1). The green scatter shows *in situ* water vapour measurements from Pico-STRAT Bi Gaz for the corresponding flight. In each figure, the red line corresponds to the slope of green scatters. It can be seen that the vertical gradient from the

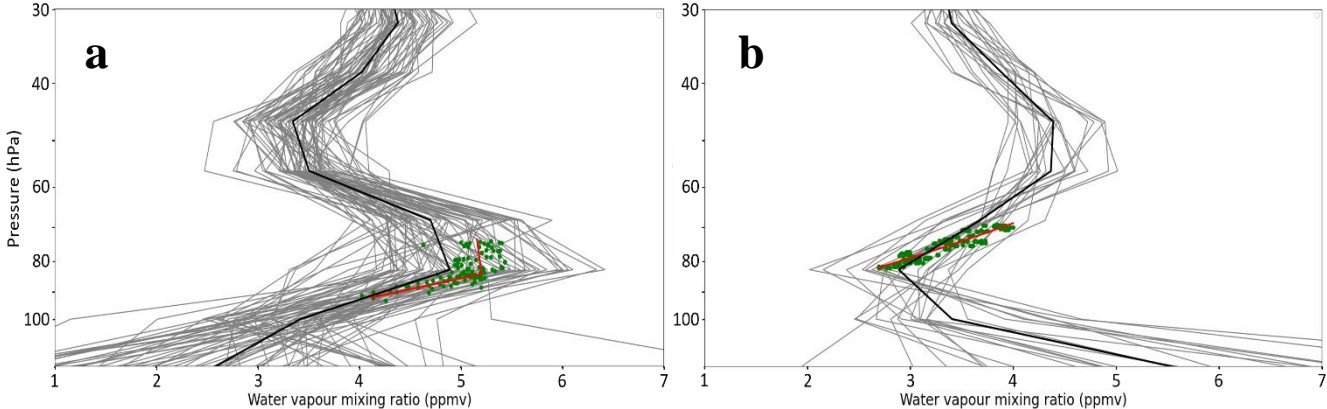

**Figure 6: Water vapour vertical profiles of MLS v5 (grey lines) taken ± 2 days around the position of depressurisation events of Pico-STRAT Bi Gaz (flight C1_07_TTL4 (11/08/2021 and 11/13/2021), in panel a) and flight C0_05_TTL2 (01/28/2020), in panel b)) and the corresponding mean MLS profile (black line). Green scatters are the Pico-STRAT Bi Gaz water vapour profiles obtained during the depressurisation (drop and rise). Red lines show the linear interpolation of the Pico-STRAT Bi Gaz profiles.**

mean MLS profiles (black line) reproduces well the *in situ* vertical gradient (red line). For one case (Fig. 6a, 11/08/2021,11/13/2021) it is interesting to see that both *in situ* and mean MLS profile capture the inversion of vertical gradient, occurring at 82 hPa.

### 3.4. Temperature products

Onboard each of the balloons, air temperature is measured *in situ* by two different instruments. Pico-STRAT Bi Gaz, includes two fast-response Sippican thermistors with an uncertainty of 0.2°C, located at each end of the optical cell. The air temperature





is measured during each acquisition of the *in situ* atmospheric spectra (every 4 to 12 minutes). Each temperature measurement
is the average of 20 readings made during 1 ms, with outliers removed. The time between measurements is sufficiently short
so that successive measurements during a flight differ by less than 0.05°C. Below the ZEPHYR gondola, the Thermodynamic
Sensors (TSEN) is a 125-µm diameter temperature sensor which performs temperature measurements every 30 seconds. Both
Pico-STRAT Bi GAz and TSEN measurements agree to $0.07\pm 0.41$ K and the Pearson's r correlation coefficient is of $0.82 \pm$
0.03 Similar agreement is found considering the ECMWF ERA5 temperature field with Pico-STRAT BI Gaz. The agreement
between the Sippican temperature measurements and ECMWF ERA5 temperature products is of around $0.435 \pm 1.38$ K and
the Pearson's r coefficient is on average of $0.74 \pm 0.09$, close to found correlations between both *in situ* sensors. A larger
standard deviation is observed in the comparison between the Sippican and ECMWF temperatures (1.4 instead of about 0.4).
Indeed, in some cases, differences as high as 8K can be observed when in situ temperature anomalies present are large
variability over rather short time periods (less than 2 days). Those cases are related to water vapour anomalies induced by
small scale atmospheric waves or deep convective events which are not well resolved in reanalysis. Best correlation is found
for the flight C1_15_TTL4 (Pearson'r = 0.82) for which the mean absolute difference between the Sippican and ECMWF
temperatures is of $0.42 \pm 1.63$ K. On the contrary, the flight C1_03_TTL4 is characterized by a Pearson's r of 0.61 while the
lowest mean difference between the Sippican and ECMWF temperatures is the lowest ($0.23 \pm 1.26$ K).

The difference in the Pearson' r coefficient between the flights C1_03_TTL4 and C1_15_TTL4 is largely influenced by the
range of probed temperatures. In the case of C1_03_TTL4, the range of probed temperature spreads from 194.7 K to 201.75 K
while the range of the C1_15_TTL4 flight extends from 189.70 K to 203.6 K. Then, the Pearson'r increases with the range of
probed temperatures. The temperature range is related, to some extent, to the amplitude of the atmospheric waves encountered
during the flight. The Pearson's r coefficient variability gives then an indirect insight on the dynamics experienced by the
instrument, especially in the case of atmospheric wave influence.
The ECMWF temperatures therefore compare well with *in situ* temperatures and can reliably be used for the analysis of wave
influence, as proposed in the following.

## 4. Results and discussion

### 4.1. Influence of atmospheric waves

In this section, we explore the impact of atmospheric waves on the modulation of water vapour by studying the correlation
between the anomalies in in situ water vapour and those in ERA5 temperature. ERA5 3D temperature field is used to build
Hovmöller diagrams, which help bringing to light the planetary- and large-scale wave activity in the vicinity of the balloons'
position. The horizontal resolution of ERA5 temperature fields is of 0.25°x 0.25° (about 30 km), thus limiting the spectrum of
atmospheric waves, which are reproduced in the reanalysis, to wavelengths greater than ~ 200 km. Therefore, the Hovmöller
diagram will not permit to identify waves of smaller wavelengths.





Figure 7 (a, b, c, d, e) shows longitude/time quasi-Lagrangian Hovmöller diagrams of temperature anomalies for the 5 flights of Strateole 2 carrying the Pico-STRAT Bi Gaz instrument. Here, the latitude range chosen to compute ERA5 temperature anomalies depend on the balloon position, i.e. the temperature anomalies are calculated hourly as the difference between the ERA5 temperatures averaged over $\pm 5°$ around the mean latitude of the balloon for each night ($T$) and the zonal mean

**Figure 7: Longitude/time "quasi-Lagragian" Hovmöller diagrams in temperature anomalies for each flight with their night-time trajectory color-coded as a function of their water vapour anomalies. (a) C0_05_TTL2. (b) C1_12_STR4 flight. (c) C1_03_TTL4 flight, (d) C1_07_TTL4 flight, (e) C1_15_TTL4 flight.**

temperature over the same latitude band ($\bar{T}$):





$$\Delta T = T - \bar{T} \tag{1}$$

In the above equation, the ERA5 temperature field are averaged on the pressure levels encompassed by each balloon. As a consequence, the pressure range and the latitude range of each graphic in Figure 7 vary from one flight to the other. Superimposed to each diagram is the trajectory of the balloon color-coded with water vapour anomalies. Figure 7 highlights

that temperature perturbations induced by atmospheric waves have an impact on the measured local water vapour anomalies. In some cases, both anomalies obviously evolve in phase during several days (see for example panel a).

In order to estimate the modulation of water vapour due to atmospheric waves, we will first calculate the Pearson's r correlation between the *in situ* water vapour anomalies and the corresponding ERA5 temperature anomalies obtained from the "quasi-Lagrangian" Hovmöller diagrams. The following development is aimed at showing that this correlation critically depends on

the vertical gradient of the water vapour mixing ratio, which is essentially controlled by the tape recorder signal in the tropical lower stratosphere. Figure 8 shows, with a green line, the wave-induced isopycnic vertical displacement of a superpressure balloon, and with red dashes, the associated isentropic vertical displacement of air masses. The water vapour anomalies measured by the balloon is thus:

$$X_b' = X(A) - X(B) = X(\bar{z}_{balloon} + \zeta_b' - \zeta_a') - X(\bar{z}_{balloon}) \tag{2}$$

where X is the water vapour mixing ratio, $\acute{z}_{balloon}$ is the balloon mean altitude, $\zeta_b'$ its vertical displacement and $\zeta_a'$ the wave-induced vertical displacement of the air mass sounded by the balloon (i.e Figure 8). Now if we develop $X(A)$ to the first order, equation 2 leads to:

$$X_b' = X(A) - X(B) = X(\bar{z}_{balloon}) + (\zeta_b' - \zeta_a') \times \frac{\partial \bar{X}}{\partial z} - X(\bar{z}_{balloon}) \tag{3}$$

where $\frac{\partial \bar{X}}{\partial z}$ is the background water vapour vertical gradient. According to Podglajen et al. (2014) $\zeta_b'$ is given by:

$$\zeta_b' = \alpha \times \zeta_a' \text{ with } \alpha = \frac{g/c_p + \partial \bar{T}/\partial z}{g/R_a + \partial \bar{T}/\partial z} \approx \frac{1}{3} \tag{4}$$

Hence, the water vapour anomaly becomes:

$$X_b' = -(1 - \alpha) \times \frac{\partial \bar{X}}{\partial z} \times \zeta_a' \tag{5}$$

Note that (1-alpha) is positive in the above equation. On the other hand, the Eulerian wave-induced temperature perturbations estimated in ERA5 are:

$$T'(A) = -\left(\frac{g}{c_p} + \frac{\partial T}{\partial z}\right) \times \zeta_a' \tag{6}$$

Hence, atmospheric waves induce a correlation between anomalies in balloon-borne observations of water vapour mixing ratios and anomalies in reanalysis temperatures. The sign of this correlation depends on the background vertical gradient of water vapour mixing ratio (hereafter VGWV) at the balloon flight level: it is positive when the gradient is itself positive, and negative otherwise. For instance, the C0_05_TTL2 flight essentially evolved in a positive vertical gradient of water vapour

mixing ratio (see Figure 2b), and the correlation between anomalies in water vapour and those in temperatures are mostly positive (see Figure 7a), as previously noted.





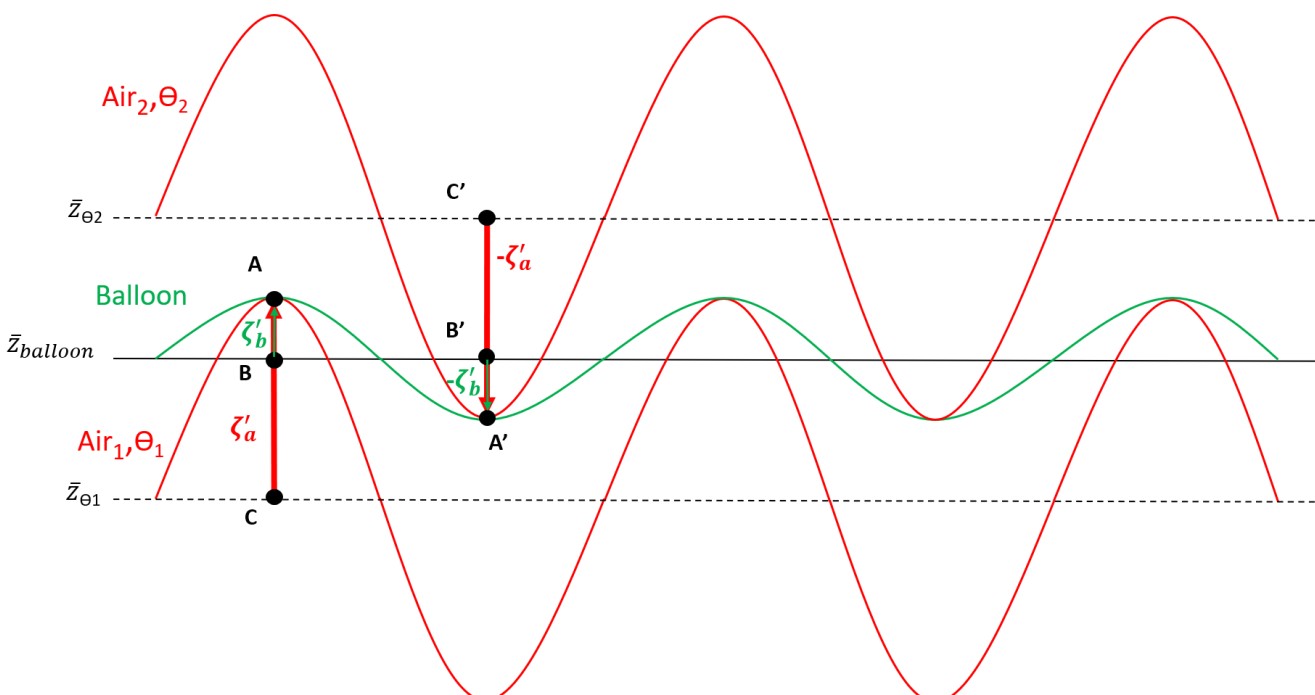

**Figure 8: Vertical displacement of the balloon (isopycnic) in green line. Vertical displacement of the airmasses due to atmospheric waves (isentropic) in red lines. The dots C and C' are the mean position of airmasses Air$_1$ and Air$_2$ at the altitude of the iso-theta 1 and 2 ($\bar{z}_{\Theta1}$, $\bar{z}_{\Theta2}$) respectively without vertical displacements. The dots B and B' are the mean position of the balloon ($\bar{z}_{balloon}$) without vertical displacements. The dots A and A' correspond to the airmass sounded by the balloon under the influence of an atmospheric wave. $\zeta'_a$ and $-\zeta'_a$ are the upward (from C to A) and downward (frome C' to A') vertical displacements of the airmasses Air$_1$ and Air$_2$ respectively due to atmospheric waves. $\zeta'_b$ and $-\zeta'_b$ are the upward (from B to A) and downward (from B' to A') vertical displacements of the balloon. The black dashed lines are respectively the mean iso-theta 1 and 2 altitude. The black line corresponds to the balloon mean altitude ($\bar{z}_{balloon}$).**





The highest correlation coefficient is actually found for the C0_05_TTL2 flight, since this flight took place in a constant positive VGWV. In such, the C0_05_TTL2 flight can be considered as a reference case, where the influence of atmospheric waves is directly highlighted by the large correlation coefficient. Regarding the other flights, the balloons flew at pressure levels often closer to the inversion of the VGWV (C1_03_TTL4 and C1_07_TTL4) or at a pressure level where the vertical gradient is negative (i.e. flight C1_03_TTL4 and C1_12_STR4), in opposition with flight C0_05_TTL2. This is further

illustrated on Figure 9 which shows the calculated MLS water vapour vertical profile over the Maritime continent covering the period from December 10, 2021 to December 18, 2021. All of the 2021 fights overflew the Maritime Continent and two (C1_07_TTL4 and C1_15_TTL4) during these 8 days. During this period, the vertical gradient reverses at 69 hPa.

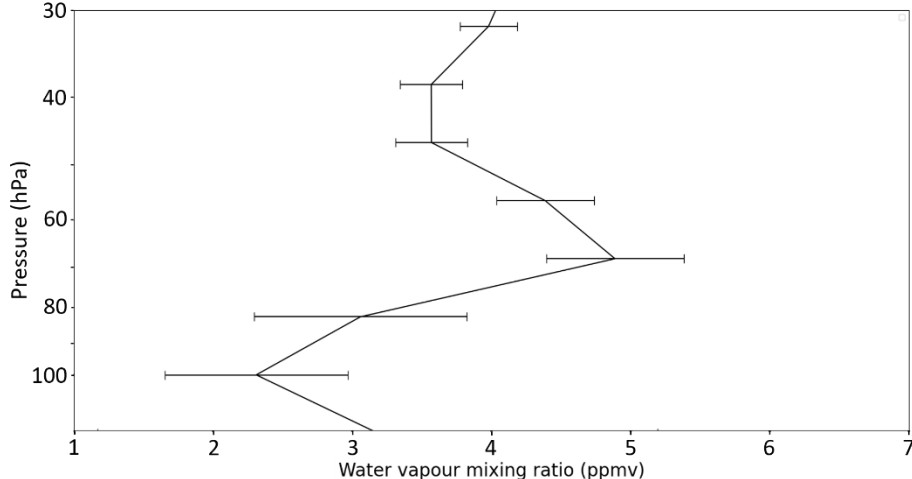

**Figure 9: Mean MLS water vapour vertical profile calculated above maritime continent with a time range of 8 days (10/12/2021 - 18/12/2021). The error bars show the standard deviation for each pressure level.**





| Flight | C0_05_TTL2 | C1_03_TTL4 | C1_12_STR4 | C1_07_TTL4 | C1_15_TTL4 |
|---|---|---|---|---|---|
| Correlation coeff. Quasi-Lagrangian Hovmöller | 0,64 | -0,39 | -0,22 | 0 | 0,31 |

**Table 2: Correlation values between "quasi-Lagrangian" Hovmöller temperature anomalies and flight water vapour anomalies**

A negative correlation, such as for the C1_03_TTL4 and C1_12_STR4 flights, does not rule out the signature of atmospheric waves in the water vapour modulation. This can be confirmed by computing the correlation between ERA5 temperature anomalies and *in situ* air temperature observed by Pico-STRAT BI Gaz. They are high for all the flights: the mean correlation coefficient is $0.77 \pm 0.06$ and individual temperature/temperature correlation coefficients are reported in Table 3.

During the period from December 28, 2021 to January 9, 2022, the C1_15_TTL4 balloon flew in a similar structure of the tape recorder as the C0_05_TTL2 balloon (i.e. positive vertical gradient, cf. Figure 2). One can thus expect similar correlation coefficient as of C0_05_TTL2 during this period for the C1_15_TTL4 flight. Indeed, restricting the calculation of the Pearson'r coefficient to the period from December 28, 2021 to January 9, 2022, leads to a correlation coefficient of 0.65, very similar to the one obtained for C0_05_TTL2. In both cases, the correlation between temperature and water vapour

anomalies therefore stands in the 0.6-0.7 range, which is the highest values observed in the Strateole-2 flights so far. Modulations of this correlation coefficient can be caused by "unfavourable" tape recorder dynamics (leading to neutral correlations) and/or additional contributions from other short time or local processes like overshooting deep convection.

| Flight | C0_05_TTL2 | C1_03_TTL4 | C1_12_STR4 | C1_07_TTL4 | C1_15_TTL4 |
|---|---|---|---|---|---|
| Correlation coeff. Quasi-Lagrangian Hovmöller | 0,79 | 0,72 | 0,71 | 0,77 | 0,85 |

**Table 3 : Correlation coefficient between Hovmöller "quasi-Lagrangian" diagram in temperature anomalies values and *in situ* temperature of the flight.**

For both C1_03_TTL4 and C1_07_TTL4 flights, the flight-mean vertical gradient is similar (i.e. 0.42 ppmv/km and 0.56 ppmv/km respectively) while in the case of C1_15_TTL4, the vertical gradient is 3 times larger (1.4 ppmv/km). These variations in the VGWV are reflected in the correlation coefficients shown in Table 2. The pair of flights C0_05_TTL2 and C1_15_TTL4 has a strongly linear correlation of water vapour anomalies with temperature since the vertical gradient is high and that the vertical displacement is large.

Besides this, one may also note that the the range of temperature (as well as potential temperature) experienced during the C1_03_TTLA and C1_07_TTL4 flights is smaller than the one experienced in C0_05_TTL2 and C1_15_TTL4 (~10 K instead of 25 K respectively) , which indicates that the wave amplitudes encountered by the former flights were smaller than for the latter ones.

A further statistical analysis was performed to estimate the relative contribution of atmospheric waves in the observed

water vapour anomalies. By computing the mean Hovmöller temperature anomaly for each night, we can determine whether





the balloon is in presence of a downward (positive temperature anomaly) or upward (negative temperature anomaly) wave-induced vertical displacement. In the Lagrangian formulation, the wave-induced displacement of air masses are isentropic such that an increase of temperature in the order of 1 K correspond to a vertical displacement of typically 100 m downward, considering the dry adiabatic lapse rate $-g/C_p = -9.8$ K.km$^{-1}$ (Note that Eulerian temperatures anomalies are actually estimated in ERA5, but the additional $-\frac{\partial T}{\partial z}$ contribution (see Eq. 6) only slightly modifies this estimate). Therefore, the mean Hovmöller temperature perturbation is associated to a "theoretical" vertical displacement. This theoretical displacement allows us to compute the corresponding air mass initial pressure level.

In order to estimate the water vapour anomaly induced by the vertical displacement, we compute a mean MLS vertical profile from MLS water vapour measurements within a 400 km radius and a few days (from 1.5 up to 10 days) around the mean balloon position. The choice of the temporal range for the calculation of the mean MLS profile does not significantly influences the statistics in the range from 1.5 up to 10 days around the mean overnight balloon position: the vertical gradient is nearly constant over this time interval:

Knowing the actual balloon mean pressure level for the given night and the theoretical vertical displacement, we can then estimate a "theoretical" water vapour anomaly. By comparing the signs of the Hovmöller temperature anomalies and of the calculated water vapour anomalies, it is possible to deduce whether the *observed* water vapour anomalies are consistent with the theoretical vertical displacement. The statistics of nights consistent with the theoretical displacement is the largest for flight C0_05_TTL2 (71% on average), in line with the high correlation coefficient (0.64). The C0_05_TTL2 flight is therefore largely influenced by atmospheric waves. In the case of the flight C1_15_TTL4, 60% of the nights are consistent with the influence of atmospheric waves though the Pearsons'r coefficient over the whole flight is of 0.31, due to the changing dynamics of tape recorder and to the influence of the typhoon Raï (as will be shown afterwards). Indeed, the C1_15_TTL4 flight has evolved almost half of time for each phase while the C0_05_TTL2 flight has evolved more than 75% of the time in the same phase, leading to a strong change in the Pearson's r correlation.

For the flights C1_07_TTL4 and C1_03_TTL4, sensitive difference in the large-scale dynamics led to smaller statistics in the number of nights influenced by waves. For both flights, the correlation coefficient is quite small, with absolute values of Pearson'r lower the 0.4 and statistics of night consistent less than 60%.

## 4.2. Impact of deep convection

The case of flight C1_15_TTL4 brings to light other sources of modulations of water vapour. From December 11, 2021 to December 16, 2021, a large and long-lasting anomaly of water vapour (close to 1 ppmv) has been detected above Papoua New Guinea and the west part of the North Pacific Ocean (see Figure 2e). In this case, the measurements were influenced by extremely deep convection associated with cyclone Rai. Considering the portions of flight which include the




influence of Rai significantly decreases the anomaly correlation described in the previous section down to 0.31, instead of 0.65 when only positive WGWV time periods are considered.

Deep convection can modulate the water vapour in two different ways. Either deep convection brings ice crystals in

the lower stratosphere which then sublimate, thereby humidifying locally the stratosphere. Or the air around the convection top is already supersaturated and the injected ice crystals can grow by solid condensation of the ambient water vapour. When large enough, the ice crystals sediment, locally drying the air. Due to the small spatial scale of overshooting tops, the hydration or dehydration signature of overshooting deep convection is expected to be detected over small areas and for short time periods (typically few hours). Though the balloon is unlikely to have flown exactly at the same time and the same location as an

overshoot, it is more likely that the balloons have flown in an air mass that was hydrated or dehydrated earlier by an overshoot.

Another possible signature of deep convection in the water vapour anomalies is the vertical displacement of isentropes due to deep convection just below (isentropic level are moving upward) or upwind (the isentropic levels are moving downwards). This behaviour is highlighted in cloud resolving model simulations of deep convection (see for example Fig. 9 in Liu et al., 2010). The corresponding signature would be the same as for atmospheric waves without freezing/drying.

While passing over deep convective systems, the balloon often decreases in altitude due to changes in infrared fluxes (depressurization). Often, before and after a depressurization, the water vapour anomalies, whether dry or wet, are large. We thus use here depressurization cases as a proxy of deep convection flyover.

Figure 10 (a, b, c) show scatter plots of water vapour anomalies as a function of the potential temperature for the three flights that have experienced depressurizations. Please note that the measurements during the balloon depressurizations are

not used to compute water vapour anomalies because it would not be possible to compute anomalies associated to a fast variation of altitude.

Most of the shape of the scatter plots can be explained by the change in time of the tape recorder signal (see figure 10). The C0_05_TTL2 and the C1_07_TTL4 flights exhibit a quasi linear shape, while the C1_15_TTL4 flight exhibits a S-shape explained by the change in time of the tape recorder state during three different periods of the flight. A contrasted

situation is observed for the flight C1_07_TTL4, for which, the water vapour vertical gradient is much smaller than for flights C0_05_TTL2 and C1_15_TTL4. In this case, the repartition of anomalies with potential temperature is significantly different, with the almost absence of a linear trend.





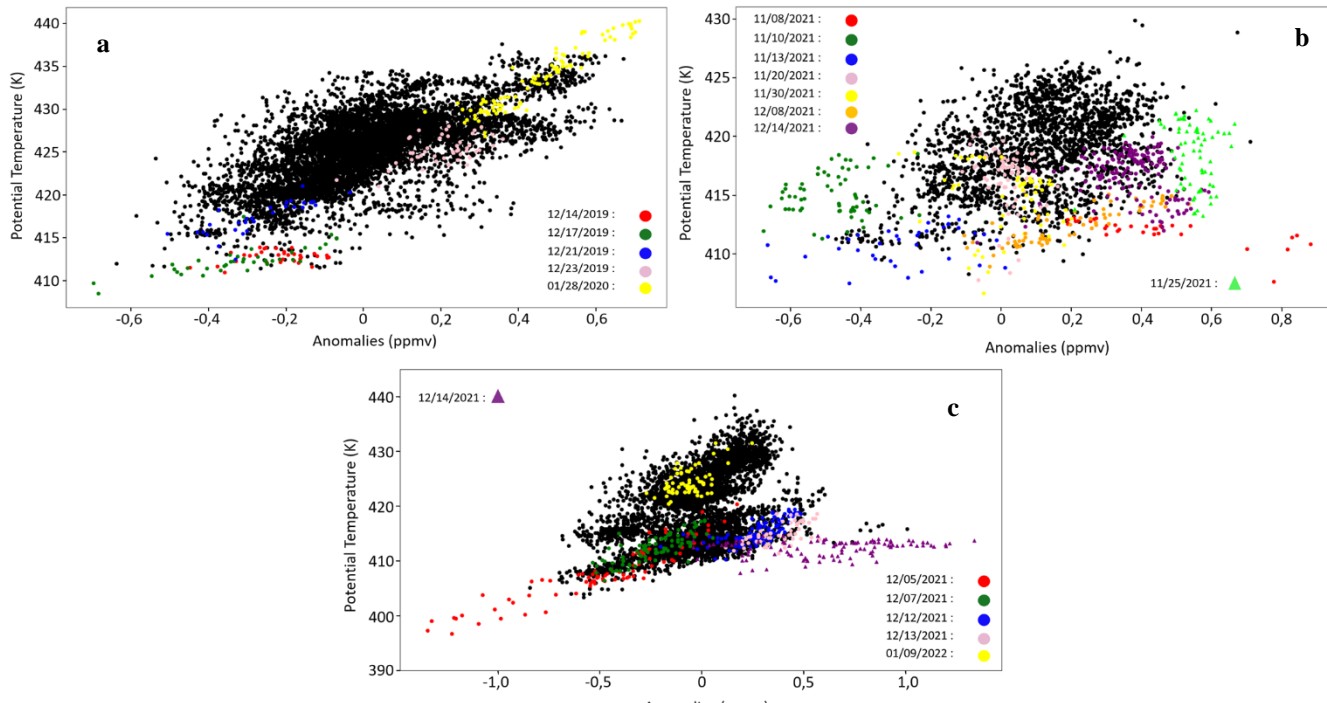

**Figure 10 : Scatterplot of water vapour anomalies as a function of potential temperature for three flights that experienced depressurizations. a) C0_05_TTL2, b) C1_07_TTL4, and c) C1_15_TTL4. For each panel, cases of depressurization are indicated by a color code. For panels b and c, specific colored triangles are indicated for nights without depressurization but for which deep convection can play a role (see text for details).**

435    In Fig. 10, the nights with depressurization are color-coded while the rest of the scatter plot data are in black. In the

latter case, it does not exclude a possible influence of deep convection.

### 4.2.1. Isentropic level vertical displacement due to local deep convection

    A linear behaviour with potential temperature during the coloured nights would indicate that water vapour variations

440 are mainly dominated by the displacement of the balloons in the vertical gradient of water vapour even in the vicinity of deep

convection. On the contrary, injection of ice by overshooting convection followed by sublimation would imply departures of

water vapour anomalies from the linear scatter plot. In addition, freezing/drying mechanism due to waves should also induce

dryer signal than the usual linear profile, as well as drying overshooting convection in supersaturated environment.

    Looking at nights when balloons have undoubtedly flown over severe deep convection (that is depressurization) no

445 systematic signature is found. Some data are especially dry: December 14, December 17 and December 21, 2019 for

C0_05_TTL2; November 10 and 13, 2021 for C1_07_TTL4; December 5, 2021 for C1_15_TTL4. But the observed water



vapour anomalies are at least qualitatively consistent with wave activity highlighted in the quasi-Lagrangian Hövmoller diagrams and the mean MLS water vapour profiles. The only exception for the dry cases is the night of November 13, 2021 for the C1_07_TTL4: the Hovmöller diagram for this night indicates a very week wave signature (very weak cold temperature anomalies), incompatible with the amplitude of the dry anomalies. The vertical displacement of isentropes due to convection just below could be an explanation since upward vertical motion would bring lower mixing ratios from below. Another possible explanation would be the impact of a small wavelength gravity wave generated by the deep convective system below that is not resolved in ERA5, so not seen in the Hovmöller diagram.

Another depressurization night of the C1_15_TTL4 flight should be underlined. Though very dry, water vapour anomalies observed on December 5 remain mostly linear with potential temperature. Such variations might be due to a wave activity superimposed to a previous wave-induced dehydration. A radiosonde from Bintulu (west of Borneo island) indicates saturated layers just above the tropopause at 00 UT on December 5, 2021. Though humidity data from radiosonde in this range of altitude should be taken cautiously, it indicates a possible supersaturation of the lower stratosphere in the region on the same day. It has to be noted that the relative humidity with respect to ice (RHI) computed from the Pico-STRAT Bi Gaz *in situ* $H_2O$ mixing ratio during that night was the highest of all the measurements during the Stratéole 2 campaign. (RHI =91.1 %)

On the opposite, several nights with depressurization events are associated with wet anomalies, C0_05_TTL2 December 23, 2019 and January 28, 2020; C1_07_TTL4 November 8 2021, December 8 and 14, 2022; C1_15_TTL4 December 12 and 13 2021. None of them shows undoubtedly a signature of deep convection because they remain in the core of the scatterplot variability, for which further analysis is required, but it is worth noticing that some distribution of the above mentioned depressurizations are at the edge of the scatterplot distribution.

In the case of flight C0_05_TTL2 on January 28, 2020 (yellow data on Fig. 10 a), the water vapour anomalies follow a linear relation with potential temperature. On the same day, an overpass of CALIOP four hours later and east of the balloon position showed deep convection that overshot the tropopause. Other cloud top products indicated very high convective cells west and four hours earlier than the balloon $H_2O$ maximums. Though the shape of the vapour data on January 28, 2020, are not typical of overshooting signature, the influence of overshooting deep convection on that day cannot be ruled out and will be the topic of another study.

For the C1_07_TTL4 flight, the wettest dots of the November 8, 2021 will be discussed in the next subsection. As for the December 8, 2021 night, though the scatter-plot shape is linear, the wettest dot cannot be compatible with a ERA5-resolved wave activity because the Hovmöller temperature anomalies are cold, and the water vapour vertical gradient is such that such a wave pattern should be associated with a dry $H_2O$ anomaly.

For the C1_15_TTL4 flight, all specific depressurization data are compatible with wave-induced or convective-induced isentropic vertical displacements, except in the case of December 12 and 13, 2021 (see following subsection). In these cases, the amplitude of the potential temperature displacement (as resolved in ERA5) is small which suggests that overshooting deep convection could play an important role.





### 4.2.2. Direct injection from deep convection


Figure 11 shows four examples of convective cases that might have influenced the water vapour anomalies and highlights the influence of the Rai tropical storm (becoming a cyclone) for two flights. Figure 11 shows cloud top products from the Himawari geostationary satellite with the corresponding balloon trajectories color coded with the water vapour anomaly. The time and date of the cloud top image is chosen to be the closest to the time when the water vapour anomaly is

the highest. Cloud top products are made available on the ICARE database. The two first panels of Figure 11 illustrate the influence of cyclone Rai, on December 13-14, 2021. First, the C1_15_TTL4 flight flew in the vicinity of a spiral rain band of Rai, relatively close to the eye, as shown in panel a) for December 13, 2021 at 18:20 UT. The online Lagrangian model HYSPLIT (Stein et al., 2015) was used to compute a back trajectory (green crosses in panel a) initiated from the wettest anomaly position and time (December 14, 2021 at 10:00 UT and the position 144.95° E, 5.3°N, and 18.6 km altitude).

HYSPLIT was fed with the GFS (Global Forecast System) analyses. This trajectory shows that the air mass with the wet anomaly, sampled by Pico-STRAT Bi Gaz, flew over the spiral rain band of Rai a few hours before the measurements. The indigo dot indicates the position of the airmass at the time of the Himawari image. The influence of Rai during this period is undeniable in Fig. 10c for December 14, 2021 (purple triangle) with a signature of quasi constant potential temperature (413 K) with a local enhancement of water vapour anomaly reaching 1.5 ppmv. It is worth noticing that the signature of December

13, 2021 in this panel may also be under the influence of Rai, with anomalies above 0.5 ppmv and out of the main scatterplot at 414 K. The balloon position close to Rai's eye on December 12, 2021 induced a depressurization. Though on the edge of the scatter plot, the water anomaly distribution is linear (blue dots) and not typical of an overshoot signature. The second flight influenced by Rai on December 14, 2021 is C1_07_TTL4. Fig. 11b shows part of the balloon trajectory and position (green dot) on December 14, 2021 at 13:20 UT when above New Guinea and the south eastern edge of Rai. The corresponding

signature in the water anomaly/theta distribution highlights a few records at 415 K out of the main scatterplot compatible with an overshoot signature, reaching almost 0.6 ppmv of water vapour anomaly. Thus, Stratéole 2 is also an opportunity to study the stratospheric hydration associated with tropical cyclones, which will be the topic of a forthcoming publication.

Besides Rai influence, the case of November 8, 2021 for C1_07_TTL4 (Fig. 10b, red scatter) departs from the main scatterplot behaviour (reported anomaly around 0.8 ppmv). During that night, the Hovmöller shows a cold wave perturbation,

which should be associated with a dry water vapour anomaly. Overshooting convection could be a possible explanation for this strong wet signature. Figure 11c shows the trajectory of the C1_07_TT4 balloon on November 8, 2021 with the corresponding water vapour anomalies superimposed on cloud top products from Himawari at 12:40 UT. It shows that on that day, south west of Sumatra, the balloon has flown over a very high convective system reaching at least 16.5 km. The resolution of Himawari (2 km x 2 km) is to coarse to detect all the overshoots reaching the stratosphere which are typically at a km$^2$ scale

but such a high main convective system anvil is compatible with overshoot within it.

By contrast, the cases which are situated at the edges of the scatter plots in Fig. 10 and are furthermore anti-correlated with what is expected from the quasi-Lagrangian Hovmöller diagram, are limit cases to the present methodology. A few



examples of such cases are summarized below. They could be associated with deep convection of smaller scales compared to the two above-mentioned cases.

515       Figure 11d illustrates the convective case of December 8, 2021 for C1_07_TTL4 flight, when the balloon is south of Sulawesi Island. The corresponding $H_2O$ anomaly/potential temperature scatterplot can be seen in orange in Fig 10b. During that night, the highest water anomaly reaches 0.53 ppmv, and as stated in section 5.2.1, this behaviour is not compatible with the signature of a wave seen in the Hovmöller diagram. A 3D back trajectory using the online HYSPLIT Lagrangian model starting from the balloon position where the vapour anomaly is the highest is computed and shown with green dots (one dot

every hours). It highlights that the air mass sampled by the balloon has overflown a very high convective cell, confirming that deep convection is a plausible explanation for the high water vapour anomaly. The Himawari cloud top image time corresponds to the time the back trajectory is above the convective cell (indigo dot at 03:00 UT).

       Not seen in Fig. 11 but seen in Fig. 10b is the case of a significant hydration above 0.65 ppmv at about 420 K on November 25, 2021 (though not during a depressurization night) during the C1_07_TTL4 flight. The vertical gradient of water

vapour calculated from the MLS v5 dataset on that day is insufficient to explain the amplitude of the hydration if it were produced by atmospheric waves. The balloon has flown in the vicinity of the decreasing phase of the Paddie tropical storm North East of Autralia, but the back trajectories calculation from this case did not undoubtedly proved the overpass of a severe convective cell from Paddie.






**Figure 11 : Himawari cloud top products for four convective cases for which specific water vapour anomalies were detected. For each panel, trajectories of the balloon are superimposed with the corresponding water vapour anomaly (color coded). The white parts of the balloons trajectory correspond to daytime, a) December 13, 2021 for the C1_15_TTL4 flight at 18:40 UT. The back trajectory from HYSPLIT initiated from the wettest anomaly position and time is shown with green crosses (one cross every hour). b) December 14, 2021 for the C1_07_TTL4 flight at 13:20 UT. The green dot is the position of the balloon at that time. c) Same as b) for November 8, 2021, 12:40 UT. d) December 8, 2021 for the C1_07_TTL4 flight at 3:00 UT. The back trajectory from HYSPLIT initiated from the position and time where the anomaly is the highest is shown with green crosses (one every hours). The position of the air mass at the time of the Himawari image is shown with an indigo dot.**

To summarize, the isentropic distribution of water vapour anomaly does not always highlight signatures of deep convection, even for nights when depressurization occurred. Some complementary approaches have to be developed to separate signals from wave and from deep convection. However, a few cases were highlighted, the most obvious of which is the signature of the Rai cyclone during the C1_15_TTL4 flight and to a lesser extent, to the C1_07_TTL4 flight. This case is associated with a stratospheric hydration from 0.6 to 1.5 ppmv and is a first step to quantify the impact of cyclone on the



stratospheric water budget. Apart from this case, a few convective-compatible signatures of water vapour were underlined with anomalies ranging from 0.4 ppmv to 0.8 ppmv (e.g C1_07_TTL4 on November 8, 2021 and December 8, 2021), though further analyses are necessaries.

**5. Conclusion**

In 2019-2020 and 2021-2022, in the frame of the two first Strateole 2 campaigns, five Pico-STRAT Bi Gaz instruments were flown from Seychelles, under super pressure balloons flying at constant density level (between 18 and 20 km) for several weeks in the upper tropical tropopause layer and lower stratosphere. The instruments performed in situ measurements of water vapour at a rate between 4 and 12 minutes during these flights. The present study exposes the methodology used to quantify the modulation of water vapour by atmospheric waves and deep convective cases. This methodology is based on water vapour

anomalies calculated from *in situ* Pico-STRAT Bi Gaz instrument and Aura MLS v5 water vapour products. The described methodology allows to put to the fore the influence of atmospheric waves and extremely deep convection on the observed water vapour anomalies.

The influence of atmospheric waves, like Kelvin waves or gravity waves, has been quantified based on the Pearsons'r correlations between ERA5 temperature anomalies and Pico-STRAT Bi Gaz water vapour anomalies. In the cases for which

atmospheric waves are the predominant mechanism in the observed modulations of water vapour, the correlation coefficient reaches up to around 0.7 if the vertical gradient of water vapour is large and if the flight takes place during a given phase of the tape recorder.

This is the case for flight C0_05_TTL2. This flight is a reference case where the tape recorder dynamics produces favourable conditions in a positive vertical gradient of water vapour throughout all the flight with a high Pearson's r coefficient. In this

case, atmospheric waves are the predominant factor for the observed modulations. In other cases, a strongly negative correlation coefficient occurs when the flight takes place during a negative vertical gradient: (flight C1_12_STR4). For the other flights (C1_03_TTL4 and C1_07_TTL4) the balloons evolved during a transition phase of the tape recorder where the vertical gradient reversed, leading to smaller correlations. We could estimate statistics of the influence of atmospheric waves, taking into account the vertical structure of the tape recorder. Indeed, flight C0_05_TTL2 is the most largely influenced by

atmospheric waves with 71% of the flight consistent with a wave influence, followed by flight C1_15_TTL4 (60%). Flights C1_07_TTL4 and C1_03_TTL4 depict the same statistics (around 55%) though, in the case of flight C1_07_TTL4, the correlation coefficient is close to zero. In such case, the flight evolved during the steady phase of the tape recorder most of the time. The extent to which the correlation coefficient is affected by the tape recorder dynamics varies upon the stratospheric dynamics experienced by the balloon, though in the case of flight C1_15_TTL4, extremely deep convection from the typhoon

Raï produces additional modulations of the Pearson'r coefficient. In case the influence is mixed with deep convection, the correlation coefficient decreases to about 0.4 or below. Here, the dynamic of the tape recorder and the associated variability of the water vapour vertical gradient modulates also the correlation coefficient throughout the whole flight.



Regarding deep convection, our method enables us to discriminate two paths in the modulation of water vapour: the direct injection for overshooting tops (large scale convective system) and the modulation due to the vertical displacement of

isentropic level due to deep convection just below deep convection. In the first case, large convective systems, for which the modulation of water vapour occurs along large distances, have a relatively large impact on the correlation coefficient: this is the case of the typhoon Raï (flights C1_15_TTL4) where a 1.5 ppmv enhancement is visible at almost constant isentropic level. In case the observed anomaly is due to a vertical displacement of isentropic levels due to convection just below, the repartition of the anomalies follows a quasi-linear trend as a function of potential temperature.


**Competing interests:** At least one of the (co-)authors is a member of the editorial board of Atmospheric Chemistry and Physics

**Acknowledgements:** The PhD work of S. Carbone was supported by CNES and CNRS. Pico-STRAT Bi Gaz data were
collected as part of Strateole-2, which is sponsored by CNES, CNRS/INSU and NSF. The team is deeply acknowledged. The authors would like to acknowledge Riwal Plougonven at LMD for his helpful comments in the preparation of this work and Alyn Lambert from the MLS team for fruitful discussions around MLS data and comparisons with in situ Pico-STRAT Bi Gaz measurements.

**Data availability**

The Pico-STRAT Bi Gaz water vapour measurements are openly available from: https://webstr2.ipsl.polytechnique.fr

The MLS water vapour measurements are openly available from: https://disc.gsfc.nasa.gov

The Himawari-8 CTH products are available from: https://www.icare.univ-lille.fr

The HYSPLIT Trajectory model is available online from: https://www.ready.noaa.gov

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
