# Peer review of "Influence of atmospheric waves and deep convection on water vapour in the equatorial lower stratosphere seen from long-duration balloon measurements"

_EGUsphere, 2024_

## Referee Comment (RC1)

Review of **Influence of atmospheric waves and deep convection on water vapor in the equatorial lower stratosphere seen from long-duration balloon measurements** by Carbone et al.

This paper makes use of data from the STRATEOLE 2 campaign to assess the influence of atmospheric waves and deep convection of water vapor in the tropical upper troposphere and lower stratosphere. The authors' method relies on variations in the Pearson correlation coefficient to determine when atmospheric waves are drivers of variations in water vapor concentration. Overall, the dataset and method used here are interesting and novel and will be of interest to ACP readers.

I do have some concerns over the paper which I believe should be addressed before publication. Specifically, the readability of the paper can be greatly improved through reduction of redundancies and careful consideration of which details need to be included for the authors to get their message across. My detailed comments are below.

General Comments:
1. There are several cases throughout the paper where figures are described in detail in both the main text and the figure caption, for example, in Line 121. Removing some of these unnecessary details from the main text may help to improve the flow of the paper
2. There are some parts of the study which seem secondary (or tertiary) to the main goal of the paper and are perhaps better suited for supplementary material to not overwhelm a reader with details that may be less necessary for their purposes. For instance, section 3.2 and associated figures 4 and 5 do not add anything substantial to the focus of the study and therefore I suggest they be added to supplementary material instead.
3. Throughout the manuscript, the Pearson's r correlation is mentioned many times, but frequently with errors varying from "Pearson's r" to "Pearson'r" to "Pearsons' r", etc. Please review and fix these occurrences.
4. Throughout the paper, there seems to be a lack of "polish" to the writing, specifically when it comes to spelling, misplaced words, and sentence structure. I tried to point out a number of these in my technical corrections below, but I am sure there are quite a few that I missed. I believe the manuscript needs to be reviewed thoroughly by the authors to ensure their message is clear to readers.

Specific Comments:
1. Lines 15-16: A bit wordy for first sentence of abstract, especially "intended to fly over the..."
2. Line 18: Either include "($H_2O$)" for consistency with other gases or get rid of the chemical formulas for the other gases
3. Line 20: Add "(r)" after Pearson correlation coefficient
4. Lines 35-38: These sentences are quite redundant; suggest combining
5. Line 39: suggest changing to "the observed increase of stratospheric..."
6. Line 40: Needs citation

7. Line 43-44: consider using the oxford comma here and throughout paper for consistency
8. Paragraph starting at Line 45: Recent literature from William Randel is very relevant and should be included in this paragraph (e.g., https://agupubs.onlinelibrary.wiley.com/doi/abs/10.1029/2019JD030648 ).
9. Line 99: Suggest adding quick general description of location of Seychelles Islands
10. Line 107: what exactly is the Zephyr gondola?
11. Line 133: Sentence starting with "This allows…" needs to be reworded, maybe something along the lines of "This allows for the dramatic reduction of the optical path length and thereby enlightens the instruments."
12. Line 139: Please define HITRAN
13. Line 148-149: This sentence is a bit unclear, suggest rewording and using a word other than "deported", perhaps "transported"?
14. Line 150-151: This sentence needs to be reworded but I also don't really understand what its trying to say. For instance, what is the compromise that is being discussed?
15. Line 157: Please include the MLS layer vertical spacing, i.e. X levels per decade of pressure
16. Paragraph starting at Line 166: I don't think the few uses of "CTH" warrant using an acronym rather than just spelling out the words
17. Figure 3a: I really love the design of this figure and think it's a very interesting way to show the comparisons of the datasets, however, the color range is a bit narrow so it is hard to fully see the differences within each MLS circle. Perhaps look at some alternative possible color tables? And ensure they are colorblind friendly
18. Tables 1-3: Why are these commas instead of periods?
19. Section 3.3: I really like this section! Figure 6 is a very pleasing result in my opinion
20. Line 203: First usage of ECMWF and ERA5, please define.
21. Figure 6 caption: Please indicate what the gray lines are
22. Section 3.4: This section was very hard for me to understand, even after reading it several times. I suggest reworking and potentially condensing as much as possible. A couple of things that I feel could benefit from clarification:
    a. What are the plus or minus ranges for all these values? A SD? 95% confidence?
    b. Be extra clear about what specific datasets are being used to calculate each pearson r. I was confused if we were comparing between observations or between one observation (which one?) and ERA 5?
    c. Speaking of ERA5, as this plays a large role in this analysis, this should be discussed/described in the data section, with a citation.
    d. Because of some of my confusion with understanding what datasets are being compared, I also don't fully understand how the Pearson r values are compared with the mean difference values and how that relates to the claimed physical mechanisms? Is it about what can be resolved in ERA5 versus what cannot? What size waves?
23. Lines 297-299: Suggest rewording to "…. Thus limiting the spectrum of atmospheric waves reproduced by the analysis to wavelengths…"

24. Line 303: The range of latitudes averaged over in ERA 5 should be added to the figure caption instead of this sentence
25. Figure 7: Think black outlines around the circles in this figure could help to make them more visible
26. Lines 312 – 336 and figure 8: This detail may be better suited for the methods section or supplementary materials, as it feels more like a methodology than results and distracts from the results you are trying to describe in this section
27. Line 360: What exactly are the "unfavorable" tape recorder dynamics?
28. Line 406: a brief discussion/description of Typhoon Rai could be helpful here for context
29. Line 463: Reword "none of them shows undoubtedly a signature of deep convection"
30. Line 523: Reword "Not seen in Fig. 11 but seen in Fig. 10b"
31. Line 526: By "vicinity of the decreasing phase", do you mean the dissipating phase of the tropical storm?
32. Line 526: Some context about tropical storm Paddie could be helpful here
33. Figure 11: I am having a very hard time reading this figure. The combination lower resolution and higher resolution cloud tops within each panels is very confusing. Why are there seemingly multiple resolutions from one product? Perhaps smoothing could help to make this figure more digestible. Additionally, I suggest making the balloon observations a bit larger so they can stand out against the background.
34. Line 545: Specify tropical cyclones, not just cyclones in general

Technical Corrections:
1. Line 31: In the light of → in light of
2. Line 40: and to the → as a result of
3. Line 45: in tropical tropopause layer → in the tropical tropopause layer
4. Line 94: accent should be removed from STRATÉOLE
5. Line 103: It is a bit hard to tell in this formatting, but there are some off occurrences of new paragraphs that should be double checked
6. Line 114: Add comma – altitude of 20.5 km thus above → altitude of 20.5 km, thus above
7. Line 127: an heritage → a heritage
8. Line 152: allows to demonstrate → demonstrates
9. Line 158: In the frame of the present study → In the present study
10. Line 271: ZEPHYR → Zephyr for consistency with the rest of the paper
11. Line 278: 8K → 8 K
12. Line 278: present are large → present large
13. Line 300: No need to mention (a,b,c,d,e) for figure 7.
14. Line 301: Strateole should be in all caps for consistency with the rest of the paper.
15. Line 395: here you stated typhoon Raï but later just say Rai, please select the correct one and use consistently throughout
16. Line 404: has been → was

17. Line 417: isentropic level → isentropic levels
18. Line 425: to a fast → with a fast
19. Line 449: week → weak
20. Line 461: "on the opposite" needs to be reworded
21. Line 503: besides → outside of
22. Line 509: to → too
23. Line 520: hours → hour
24. Line 527: Autralia → Australia
25. Line 553: Suggest changing "exposes" to a different word
26. Line 568: could → can
27.

---

## Author Comment (AC2)

**Influence of atmospheric waves and deep convection on water vapour in the equatorial lower stratosphere seen from long-duration balloon measurements**

Sullivan Carbone1, Emmanuel D. Riviere1, Mélanie Ghysels1, Jérémie Burgalat1, Georges Durry1, Nadir Amarouche2, Aurélien Podglajen3, Albert Hertzog3

[revised manuscript text omitted]

---

## Author Response (AR1)

**Dear reviewer #1,**

**Thank you for taking time to reply to review our manuscript.**

**Please, find our response to your comments below.**

**In order to help you figure out where changes have been made in the paper, we also  attach a specific version of the revised paper changes are writter in blue (following both reviewers' request).**

**Best regards,**
* * *
Review of Influence of atmospheric waves and deep convection on water vapor in the

equatorial lower stratosphere seen from long-duration balloon measurements by

Carbone et al.

This paper makes use of data from the STRATEOLE 2 campaign to assess the influence of atmospheric waves and deep convection of water vapor in the tropical upper troposphere and lower stratosphere. The authors' method relies on variations in the Pearson correlation coefficient to determine when atmospheric waves are drivers of variations in water vapor concentration. Overall, the dataset and method used here are interesting and novel and will be of interest to ACP readers.

I do have some concerns over the paper which I believe should be addressed before publication. Specifically, the readability of the paper can be greatly improved through reduction of redundancies and careful consideration of which details need to be included for the authors to get their message across. My detailed comments are below.

General Comments:

1. There are several cases throughout the paper where figures are described in detail in both the main text and the figure caption, for example, in Line 121. Removing some of these unnecessary details from the main text may help to improve the flow of the paper.

**Authors: Done. We removed unecessary details in the text.**

2. There are some parts of the study which seem secondary (or tertiary) to the main goal of the paper and are perhaps better suited for supplementary material to not overwhelm a reader with details that may be less necessary for their purposes. For instance, section 3.2 and associated figures 4 and 5 do not add anything substantial to the focus of the study and therefore I suggest they be added to supplementary material instead.

**Author: The section 3.2 has been moved to the Appendix A1.**

3. Throughout the manuscript, the Pearson's r correlation is mentioned many times,

but frequently with errors varying from "Pearson's r" to "Pearson'r" to "Pearsons' r",

etc. Please review and fix these occurrences.

**Authors: Done. We therefore used a consistent terminalogy, which is Pearson's correlation coefficient.**

4. Throughout the paper, there seems to be a lack of "polish" to the writing, specifically when it comes to spelling, misplaced words, and sentence structure. I tried to point out a number of these in my technical corrections below, but I am sure there are quite a few that I missed. I believe the manuscript needs to be reviewed thoroughly by the authors to ensure their message is clear to readers.

**Authors: we have proceeded with wording revisions. We hope that the revised version is better.**

**Specific Comments:**

1. Lines 15-16: A bit wordy for first sentence of abstract, especially "intended to fly over the..."

**Authors: We reworded the sentence such as : « The Strateole 2 project consists of 3 campaigns of super pressure balloons released from the Seychelles. The balloons have flown in the whole equatorial belt transported by winds during 3 to 4 months. »**

2. Line 18: Either include "($H_2O$)" for consistency with other gases or get rid of the chemical formulas for the other gases

**Authors: We have added « $H_2O$ » to water vapour for consistency with other gases.**

3. Line 20: Add "(r)" after Pearson correlation coeEicient

**Authors: In the whole manuscript, we choose the denomination « Pearson's correlation coefficient».**

4. Lines 35-38: These sentences are quite redundant; suggest combining

**Authors: We revised such as : « Observational studies have shown that the global temperature is sensitive to small changes of water vapour in the lower stratosphere (Forster and Shine, 1999; Solomon et al., 2010; Wang et al., 2017). »**

5. Line 39: suggest changing to "the observed increase of stratospheric…"

**Authors: Done. We included the following citations : (Oman et al., 2008; Noël et al., 2018; Tian and Chipperfield, 2006).**

Oman, L., D. W.Waugh, S. Pawson, R. S. Stolarski, and J. E. Nielsen, 2008: Understanding theChanges of Stratospheric Water Vapor in Coupled Chemistry–Climate ModelSimulations. J. Atmos. Sci., 65, 3278–3291, https://doi.org/10.1175/2008JAS2696.1.

Noël, S., Weigel,K., Bramstedt, K., Rozanov, A., Weber, M., Bovensmann, H., and Burrows, J. P.:Water vapour and methane coupling in the stratosphere observed using SCIAMACHYsolar occultation measurements, Atmos. Chem. Phys., 18, 4463–4476, https://doi.org/10.5194/acp-18-4463-2018,2018.

Tian, W.,and M. P. Chipperfield (2006), Stratospheric water vapor trends in a coupled chemistry-climate model, Geophys. Res. Lett., 33,L06819, doi:10.1029/2005GL024675.

6. Line 40: Needs citation

**Authors: Done.**

7. Line 43-44: consider using the oxford comma here and throughout paper for

consistency

8. Paragraph starting at Line 45: Recent literature from William Randel is very relevant

and should be included in this paragraph (e.g.,

https://agupubs.onlinelibrary.wiley.com/doi/abs/10.1029/2019JD030648 ).

**Authors: Thank you for suggesting this reference. We have included it to the citations.**

9. Line 99: Suggest adding quick general description of location of Seychelles Islands

**Authors: We have slightly revised the sentence such as : « Strateole 2 relies on 3 long-duration campaigns where a flotilla of superpressure balloons is launched from the Mahé Island, the Seychelles archipelago, in the Indian Ocean off East Africa ».**

10. Line 107: what exactly is the Zephyr gondola?

**Authors: We provided additional details in the manuscript :**

**« The balloons can carry up to 15 kg of scientific instrumentation, allowing to probe several meteorological and chemical variables (wind, pressure, temperature, aerosols, clouds, water vapour, other gases etc.) in situ. The well-functioning of the instruments is ensured by the Zephyr gondola, which is located 2 meters upstream Pico-STRAT Bi Gaz on the flight chain. Zephyr is a gondola which provides power (through solar panels), positioning and timing information (onboard GPS receiver) and communication to the ground control center (using iridium space-borne communication system) to scientific**

**instruments. Some instrumentations are located inside the Zephyr gondola to protect electronics from the environment cold temperatures (e.g. TSEN). »**

11. Line 133: Sentence starting with "This allows…" needs to be reworded, maybe

something along the lines of "This allows for the dramatic reduction of the optical

path length and thereby enlightens the instruments."

**Authors: We propose the following revision : « The strong line intensities of fundamental bands allow to dramatically reduce the optical path length and thereby enlighten the instruments. »**

12. Line 139: Please define HITRAN

**Authors: The acronyme has been defined.**

13. Line 148-149: This sentence is a bit unclear, suggest rewording and using a word

other than "deported", perhaps "transported"?

**Authors: the sentence has been reworded : « The optical cell, including the laser diodes and the detectors, is not located in Zephyr but hanging down below the Zephir gondola, at a 2-m distance. »**

14. Line 150-151: This sentence needs to be reworded but I also don't really understand

what its trying to say. For instance, what is the compromise that is being discussed?

**Authors: Measuring water vapour in the dry TTL requires that the instrument lies in ambient air and not within the Zephyr gondola. The super pressure balloon capacity limits the instrument total weight to 5 kg. To protect the electronics of Pico-STRAT Bi Gaz from frost, it has been installed in Zephyr. The rests of the instrument (the optical cell : sensing area) has to hang down below by 2m. Such distance require to design a feeding cable, weighting already 1.2 kg.**

**Due to the 5kg limit, the length of the cable had to be limited to 2m, which has some impact on the measurements : daytime contamination from outgassing (from Zephyr and balloon), which could be reduced if the distance between the optical cell and the rest of the flight chain (Zephyr and balloon) was much longer (typically 30 m). Unfortunately, such length could not be implemented due to the overweight it would cause and other technical issues related to CNES certification process.**

**We revised this paragraph to clarify, such as :**

**« The optical cell, including the laser diodes and the detectors, is not located in Zephyr but hanging down below the Zephir gondola, at a 2-m distance, to limit the contamination of water vapour measurements from outgassing Zephyr and balloon surfaces. In this**

**configuration, the electronics module is connected to the lasers and the detectors using 2.5-m shielded cables. »**

15. Line 157: Please include the MLS layer vertical spacing, i.e. X levels per decade of pressure

**Authors: Done.**

16. Paragraph starting at Line 166: I don't think the few uses of "CTH" warrant using an acronym rather than just spelling out the words

**Authors: this is to slightly improve the flow.**

17. Figure 3a: I really love the design of this figure and think it's a very interesting way to show the comparisons of the datasets, however, the color range is a bit narrow so it is hard to fully see the differences within each MLS circle. Perhaps look at some alternative possible color tables? And ensure they are colorblind friendly.

**Authors: We slightly adjusted the color scale of the anomalies ($H_2O$) so the contrast between in situ and mean MLS value is more visible. It passes COBLIS tests.**

18. Tables 1-3: Why are these commas instead of periods?

**Authors: Done.**

19. Section 3.3: I really like this section! Figure 6 is a very pleasing result in my opinion

**Authors: Thank you!**

20. Line 203: First usage of ECMWF and ERA5, please define.

**Authors: we included the acronym signification of ECMWF and ERA 5 definition.**

21. Figure 6 caption: Please indicate what the gray lines are

**Authors: After moving the old section 3.3 to the Appendix, figure 6 becomes Figure 4.**

**The grey lines are the MLS vertical profiles taken ± 2 days around the position of depressurisation events of Pico-STRAT Bi Gaz. This is already stated in the caption.**

22. Section 3.4: This section was very hard for me to understand, even after reading it

several times. I suggest reworking and potentially condensing as much as possible.

A couple of things that I feel could benefit from clarification:

a. What are the plus or minus ranges for all these values? A SD? 95%

confidence?

b. Be extra clear about what specific datasets are being used to calculate each

pearson r. I was confused if we were comparing between observations or

between one observation (which one?) and ERA 5?

c. Speaking of ERA5, as this plays a large role in this analysis, this should be

discussed/described in the data section, with a citation.

d. Because of some of my confusion with understanding what datasets are

being compared, I also don't fully understand how the Pearson r values are

compared with the mean diEerence values and how that relates to the

claimed physical mechanisms? Is it about what can be resolved in ERA5

versus what cannot? What size waves?

**Authors: we revised the section, removing extra information which may have caused confusion.**

**The key elements are that : 1- the ERA5 temperature reproduce well in situ observations, and therefore are reliable for the analysis ; 2- Some largers differences between in situ Pico-STRAT and ERA5 temperatures can be observed in the case of in situ modulations due to waves or to deep convective events which are not resolved by the reanalysis.**

23. Lines 297-299: Suggest rewording to "…. Thus limiting the spectrum of atmospheric

waves reproduced by the analysis to wavelengths…"

**Authors: we revised the sentence such as : « The horizontal resolution of ERA5 temperature fields is of 0.25°x 0.25° (about 28 km), thus limiting the spectrum of atmospheric waves resolved by the analysis to horizontal wavelengths greater than ~ 120 km. »**

24. Line 303: The range of latitudes averaged over in ERA 5 should be added to the figure

caption instead of this sentence

**Authors: we revised this paragraph such as « Figure 5 shows longitude/time quasi-Lagrangian Hovmöller diagrams of temperature anomalies for the 5 flights of Strateole 2 carrying the Pico-STRAT Bi Gaz instrument. The temperature anomalies (ΔT) are calculated such as:**

$$\Delta T = T - \bar{T}$$

**Where T is the average temperature of ERA 5 over ±5° around the mean latitude of the balloon for each night and  is the zonal mean temperature over the same latitude band.$\bar{T}$ ».**

**We also completed the caption of Figure 7 (now FIgure 5) such as:**

**« Figure 5. Longitude/time "quasi-Lagragian" Hovmöller diagrams in temperature anomalies for each flight with their night-time trajectory color-coded as a function of their water vapour anomalies. The temperature anomalies are calculated hourly as the difference between the ERA5 temperatures averaged over ±5° around the mean latitude of the balloon for each night and the zonal mean temperature over the same latitude band. (a) C0_05_TTL2. (b) C1_12_STR4 flight. (c) C1_03_TTL4 flight, (d) C1_07_TTL4 flight, (e) C1_15_TTL4 flight. ».**

25. Figure 7: Think black outlines around the circles in this figure could help to make

them more visible

**Authors: We are not sure to understand what you mean. If you mean that each small circle of the trajectory of the balloon should be surrounded by black lines, it would certainly bring more contrast with respect to the temperature anomalies of the Hovmöller diagrams, but the water vapour anomalies surimposed to the balloon trajectory would then be unreadable. In some cases, the circles can overlap, especially when the balloon is turning around.**

**We therefore surimposed the balloon trajectory in black dash line to improve visibility on the trajectory location.**

26. Lines 312 – 336 and figure 8: This detail may be better suited for the methods

section or supplementary materials, as it feels more like a methodology than results

and distracts from the results you are trying to describe in this section

**Authors: We do understand your point. However, we prefer to keep this section because it directly illustrate the interpretation of the results and provides clue to the reader to understand the approach (correlation or anti-correlation between the temperature anomalies due to waves and the water vapour anomalies).**

27. Line 360: What exactly are the "unfavorable" tape recorder dynamics?

**Authors: Unfavourable tape recorder dynamics is when the balloon evolves at a level close to vertical gradient reversal or in an altitude range where the vertical gradient of water vapor is small.**

**We revised the text such as : « Modulations of this Pearson's correlation can occur when, on some portion of the flight, the balloon evolves at a level close to a vertical gradient reversal or in an altitude range where the vertical gradient of water vapour is small (leading to null correlations). Additional contributions from other short time or local processes like overshooting deep convection can also be a cause. »**

28. Line 406: a brief discussion/description of Typhoon Rai could be helpful here for

context

**Authors: we have added elements of information about Rai : « Rai is one of the most intense typhoons of the 2021 season. It started to develop on December 8, 2021, in the Northeast of New Guinea, in the Pacific Ocean. It reached the Category 1 of the Saffir-Simpson scale on December 14, 2021. On December 15-16, 2021, it reached the Category 5 and hit the Philippines. »**

29. Line 463: Reword "none of them shows undoubtedly a signature of deep

convection"

**Authors: we have rephrased the paragraph so the meaning is more clear, such as :**

**« On the opposite, several nights with depressurization events are associated with wet anomalies, C0_05_TTL2 December 23, 2019, and January 28, 2020; C1_07_TTL4 November 8, 2021, December 8 and 14, 2022; C1_15_TTL4 December 12 and 13 2021. For these cases, satellite observations of HIMAWARI show that the balloons were overpassing deep convective systems while no direct signature of direct injection has been observed. Instead, the signatures which are observed depict a quasi-linear trend with potential temperature, suggesting isentropic displacements, but the amplitude of the anomalies cannot be explained only considering those displacements. These cases represent the limit of our methodology and are not easily interpreted.»**

30. Line 523: Reword "Not seen in Fig. 11 but seen in Fig. 10b"

**Authors: Done. We rephrased such as « On November 25, 2021, a significant hydration (higher than 0.65 ppmv) at about 420 K is observed (though not during a depressurization night) during the C1_07_TTL4 flight (see Fig. 8b). »**

31. Line 526: By "vicinity of the decreasing phase", do you mean the dissipating phase of

the tropical storm?

**Authors: Yes. We therefore rephrased « The balloon has flown in the vicinity of the Paddie tropical storm Northeast of Australia, while the storm was dissipating. The back trajectories calculation did not undoubtedly prove the overpass of a severe convective cell from Paddie. »**

32. Line 526: Some context about tropical storm Paddie could be helpful here

**Authors: We have included a short description of Paddy (misspelled in the original version of the article) :**

**« The balloon has flown in the vicinity of the Paddy tropical storm Northeast of Australia, while the storm was dissipating. Paddy was a relatively short-lived storm that**

**formed North-West of Australia. It did not reach the typhoon category but impacted Micronesia. »**

33. Figure 11: I am having a very hard time reading this figure. The combination lower resolution and higher resolution cloud tops within each panels is very confusing. Why are there seemingly multiple resolutions from one product? Perhaps smoothing could help to make this figure more digestible. Additionally, I suggest making the balloon observations a bit larger so they can stand out against the background.

**Authors: To plot this figure, we are using the CTTH product made available for the scientific community. We are not able to justify why, in some specific locations and dates, the resolution of the product is degraded. So there is nothing we can do get rid of this combination of different resolutions within each pannels. We have carried out different tests of smoothing, but it didnt make any improvement. So here, we decided keep the same resolution as it was previously. However, answering to your second point, we have made the ballon observations a bit larger in each panels**.

34. Line 545: Specify tropical cyclones, not just cyclones in general

**Authors: Done.**

**Technical Corrections:**

**Authors: We have applied all the below-listed technical revisions and, in general, revised the wording of all the manuscript.**

1. Line 31: In the light of à in light of

2. Line 40: and to the à as a result of

3. Line 45: in tropical tropopause layer à in the tropical tropopause layer

4. Line 94: accent should be removed from STRATÉOLE

5. Line 103: It is a bit hard to tell in this formatting, but there are some off occurrences of new paragraphs that should be double checked

6. Line 114: Add comma – altitude of 20.5 km thus above à altitude of 20.5 km, thus above

7. Line 127: an heritage à a heritage

8. Line 152: allows to demonstrate à demonstrates

9. Line 158: In the frame of the present study à In the present study

10. Line 271: ZEPHYR à Zephyr for consistency with the rest of the paper

11. Line 278: 8K à 8 K

12. Line 278: present are large à present large

**Authors: This comment do not apply anymore following revisions of this subsection.**

13. Line 300: No need to mention (a,b,c,d,e) for figure 7.

14. Line 301: Strateole should be in all caps for consistency with the rest of the paper.

15. Line 395: here you stated typhoon Raï but later just say Rai, please select the

correct one and use consistently throughout

16. Line 404: has been à was

17. Line 417: isentropic level à isentropic levels

18. Line 425: to a fast à with a fast

19. Line 449: week à weak

20. Line 461: "on the opposite" needs to be reworded

**Authors: we used « On the other side »**

21. Line 503: besides à outside of

22. Line 509: to à too

23. Line 520: hours à hour

24. Line 527: Autralia à Australia

25. Line 553: Suggest changing "exposes" to a different word

26. Line 568: could à can

-------------------------------------------------------------------------------------------------------------------- --------------

**Dear reviewer #2,**

**Thank you for taking time to reply to review our manuscript.**

**Please, find our response to your comments below.**

**Also attached a new version of the paper with changes provided in blue, according to both referees' request**

**Best regards,**

———————————————————————————————————————————————————————-

Carbone et al. provide a study where they relate water vapor distribution with wave activity and deep convection. They employ super pressure balloons observations to give unique insights on the influence of tropical waves and deep convection on stratospheric water vapor. In their study they identify one flight with clear indication of water vapor modulation of waves, while the influence of deep convection is analyzed on the example of typhoon Rai. The paper is well organized, and the science is novel and after a few clarifications worthy of publication. I thoroughly enjoyed reading the manuscript and mostly have technical comments to improve readability.

Main comments:

Line 76f: The sentence does not make sense at the moment. Please rephrase.

**Authors: Done**

Line 85: "Water vapor anomalies…" this sentence needs to be moved to the methods section

**Authors: we have removed this sentence since it is redundant with the introductive paragraph of section 3. Methodology.**

Line 103f: I would suggest rephrasing the sentence to: once at float level, the balloons drift with the wind either east- or westward depending on the QBO phase.

**Authors: thank you for this suggestion. Done.**

Line 111: what does "has evolved in the wet phase" mean?

**Authors: We wanted to enphasize that the flight occurred within the wet phase of the tape recorder, but this was not well written. Instead, we replaced the sentence by : « It flew for 79 days within the wet phase of the tape recorder ».**

Line 365-369: I don't fully understand the reasoning here. Why are the VGWV variations reflected in the correlation coefficients? The VGWV for C1_03_TTL4 and C1_07_TTL4 are similar, however the correlations are very different (from no correlation at all to 0.39).

**Authors: We agree. This paragraph has been removed.**

Also I would be interested in why there is no correlation for C1_07_TTL4 balloon? How far apart were these balloons? In other words, why does one balloon show a correlation signal and the other does not? Are they geographically far apart? Where they flying through different convective systems?

**Authors: The calculation of a mean H₂O MLS profile nearby the balloon position and location depicts the vertical structure of H₂O. We can see an oscillation of the water vapour mixing ratio with altitude. (see figure below the mean MLS vertical profile).**

[Figure]

H$_2$O mixing ratio (ppmv)

**Our analysis shows that the flight C1_07_TTL4 took place during a phase of tape recorder where the balloon was flying close to the altitude of H₂O vertical gradient inversion (red dot on the figure above). This implies that during isentropic vertical displacement (of few hundreds meters or less) the mixing ratio does not vary significantly, while the temperature changes. This leads to the poor correlation coefficient. This configuration is similar for flight C1_03_TTL4.**

**The flights C0_05_TTL2 and C1_15_TTL4 are in the configuration of the blue dot on the figure above.**

**In the manuscript, we added a short paragraph which explains the flights' configurations.**

Line 395: Please rephrase the sentences: what does "... flight has evolved almost half the time for each phase..." mean?

**Authors: We rephrased : « Indeed, the C1_15_TTL4 flight has evolved half of time within each phase of the tape recorder (moistening/drying) while the C0_05_TTL2 flight has evolved more than 75% of the time in the same phase (drying), leading to a strong change in the Pearson's r correlation. »**

Line 398ff: I'm surprised. C1_15_TTL4 has a lower correlation coefficient in Table 2 and yet almost half the time influenced by waves if I understand correctly. So I would expect the same if not more for C1_03_TTL4. Do you have an explanation for that difference?

**Authors: this is in line with our previous reply. The Pearson's r coefficient on its own is not sufficient to conclude on the importance of atmospheric waves on our H₂O time series.**

It could be sufficient if a given flight had evolved during the same phase of the tape recorder and that the balloon was flying at a favorable altitude (not too close of the altitude where the vertical gradient reverses).

One has to consider both coefficients and large-scale dynamics (the tape recorder-induced vertical variability of $H_2O$). The statistics of the number of nights influenced by waves is determined by comparing the sign of the water vapour anomalies and the sign of the theoretical vertical gradient of $H_2O$. The correlation coefficient between $H_2O$ anomalies and temperature can decrease if a sufficient number of nights occur in a configuration where the balloon is influenced by vertical displacements in a negative vertical gradient. In this case the correlation coefficient decreases but still the $H_2O$ anomalies are coherent with the vertical gradient (and then are considered as influenced by waves).

The flight C1_03_TTL4 occurred in a steady phase of the tape recorder where the vertical gradient of $H_2O$ nearby the balloon is small. Additionally, the calculated vertical displacement are the smallest.

Line 510: I did not understand the last half of the sentence. Please rephrase. What does "… system anvil is compatible with overshoot within it" mean?

Authors: We rephrased the sentence : « The resolution of Himawari (2 km x 2 km) is to coarse to detect all the overshooting tops reaching the stratosphere which are typically at a $km^2$ scale. »

Figure 6: I would suggest to zoom into this plot and have the top at 50hPa that way it is easier to see the variation of Pico-STRAT Bi Gaz observations. Also the inversion is an important feature and therefore I would suggest to highlight it in the figure.

Authors: We have reploted the figure having the top at 40 hPa (and not 50 hPa) : it allows to see the inversion altitude.

Figure 7: Some of the circles are difficult to see, I would suggest to make them bigger.

Authors: Making the circles bigger would increase the overlap between consecutive circles, especially in cases where the balloon operated a turn around and when the winds are weak (slow motion). We therefore surimposed the balloon trajectory in black dash line to improve visibility on the trajectory location.

Figure 10: It's impossible to follow the discussion of this figure as you can only see black dots. Maybe work with different transparencies? Description of black dots is missing in the figure caption.

Authors: The figure caption has been clarified. We added the following sentence : « Black dots are for any other dates »

---

## Editor Decision (ED1)

**Editor comments on egusphere-2024-3249**

*Influence of atmospheric waves and deep convection on water vapour in the equatorial lower stratosphere seen from long-duration balloon measurements* by Carbone et al.

Although this manuscript went already through a round of revisions, there are many things left that need to be corrected before the manuscript can be accepted for publication. Thus, another round of major revisions is necessary.

**General comments:**

1. Please consider the ACP manuscript preparation guidelines and prepare your manuscript accordingly.

   https://www.atmospheric-chemistry-and-physics.net/submission.html#manuscriptcomposition

   In the abstract the introductory part and the concluding part is missing, see the specific guidelines for title, abstract and conclusions:

   https://www.atmospheric-chemistry-and-physics.net/policies/guidelines_for_authors.html

   Please revise the abstract accordingly. See also my comments given below under specific comments.

2. In the text Section and Figure abbreviated as Sect. and Fig. unless they appear at the begin of the sentence. Please correct this throughout the manuscript.
3. Paragraphs should have a reasonable length. Please check all paragraphs and combine the ones which consists only of 1-3 sentences. The text becomes hardly readable to to this from page 10 on there this has excessively been done,
4. The whole manuscript feels a bit mixed up. Text that belongs to the results is added in the methodology section and text that belongs to methodology is found in the results. Also the data is described in the wrong order. MLS data is used already before the data and instrument are described. Please sort your text and prepare the manuscript according to the manuscript preparation guidelines.

**Specific comments:**

P1, L15: An introduction to the field is missing. What is the purpose of this study?

P1, L22 and L26: Using the names you have given your flights in the abstract is meaningless for the reader. Use the dates instead.

P1, L27: The concluding part in the abstract is missing. What are the implications of your study and results?

P2, L60: Has the abbreviation "CPT" been introduced?

P2, L65: change to "that not necessarily reaches the stratosphere, see e.g. Rysman et al., 2017)

P4, L105: Don't make paragraphs consisting of one line. Add this sentence to the following paragraph.

P4, L111: What do you mean with "the environmental cold temperature"? Do you mean "cold environmental temperatures"? Please rephrase/correct the sentence.

P4, L112: What is "TSEN"? The abbreviation has not been introduced.

P4, L117ff: Using the flight names here alone is not helpful for the reader. Please add the dates and also add a table summarizing the names and dates for the flights used in your study.

P4, L122: Who is meant with "They"? The balloons? Please be more precise.

P5, L124: Who or what is meant with "it"?

P5, L126: To which figure are you referring to?

P5, L127: What do you mean with "the balloons evolved"? Please rephrase,

P5, Figure 2: MLS used here, but neither the data or instrument have been described yet. Doesn't this figure rather belong to the result section?

P6, L132: What is "SLDA"? The abbreviation has not been introduced.

P6, L138: "regions of strong absorptions within fundamental bands" not clear. Make an own sentence clearly writing what you mean.

P6, Section 2.2. (and also previous section): A large part of what is written here goes beyond the pure data description and rather belongs to the methodology section. Move these parts to the methodology section and rename this one to e.g. "Balloon flights and methodology".

P6, L193: It is not clear what kind of methodology you have developed. Is this really new? I have the feeling already known methods are used here.

P6, L197: How do you know that these are caused by overshooting convection and atmospheric waves? Provide an explanation for the impact of overshooting convection and atmospheric waves on the anomalies and also provide some references.

P7, L161: Description of MLS appears after it the data has been used. See my comment above. The sections should be revised and appear in a different order.

P7, L164-165: The sentence "The vertical grid ….." is not clear and should be revised.

P7, L171-172: Remove line break. This sentence should be added to the previous paragraph.

P7, L182: "……. and by using cloud-type data deduced from cloud-type and phase product" not clear and needs to be revised.

P7, L188: Why do you want to evaluate ERA5? Aren't you just comparing these data sets to see if these can be applied together?

P7, L188: Which products are used? List these here explicitly.

P8, L199: Combine the sentences and replace "indeed" by "since" or "because".

P8, L201: trapped in what? Clouds? Not clear. Please rephrase sentence,

P8, L205: delete "first" and replace "followed" by "based on".

P8, L205: add the criteria here (again) or refer to the previous paragraph/section where these are stated.

P9, L216: Introducing ERA5 and ECMWF comes here long after ERA5 has already been mentioned and used. This introduction should be made at the begin of Sect. 3.

P9, Figure 4: The green dots and red line are not that good visible. The bullets for the dots should be larger and the red line plotted somewhat thicker.

P10, L240: Why evaluation? This is rather just a comparison between the data sets.

P10, L241-249: This text part belongs to Sect. 2.2 where the balloon instrument should be described. Further make paragraphs with a reasonable length and not one paragraph for just one or two sentences.

P11, L266: Which data is shown here? MLS? Please clearly state which data is shown.

P11, Figure 5 caption: You mean the respective balloon trajectory with the MLS water vapour color coded?

P11, Figure 5: Which data sets have been used to derive the Hovmöller diagram is not clear. Maybe it would also be good to mention somewhere what a Hovmöller diagram is and for what it is used.

P12, L268-270: This sentence belongs to the data description section. Rephrase the sentence and refer to the respective section and only point here out the limited resolution of the ERA5 data.

P12, L274ff: The entire remainder of the page belongs to the methodology section. Also revise here the paragraphs so that each has a reasonable length.

P13, L306: add the date.

P14, L314: What is "VGWV"? The introduction has not been introduced.

P15, L327: Sentence should be added to the previous paragraph.

P15, Table 2: This table appears in the middle of the text (last part of the paragraph appears between table and table caption).

P16, L358-361: These sentences should be added to the previous paragraph.

P16, L381-382: Add this paragraph either to the previous or following paragraph.

P17, L394-395: Again one paragraph for just one sentence.

P17, L396: Which data? MLS? The balloon data?

P17, Figure 8 caption: Add that the dates for the flights are given in the figure legend.

P18, L407-408: Again one paragraph for just one sentence. This sentence also rather belongs to the figure caption than in the main text.

P18, L412: Do you mean colored in the figure? Be more clear.

P18, L412: Sentence "displacement of the balloons in the vertical gradient of water vapour " not clear. What do you mean? Another flight showing a different case? Please rephrase.

P18, L422-425: Where is this shown? To which figure are you referring to?

P18, L426: Sentence "Another case of flight C1_15_TTL4 should be underlined" not clear and should be rephrased.

P19, L435: What is HIMAWARI? The abbreviation has not been introduced.

P19, L441: The abbreviation CALIOP has not been introduced. Also it should be mentioned what kind of instruments these are.

P19, L457: Add some more details on the storm Rai (when did it form, how long did it last, from where to where did it move and what implications had this storm).

P19, L460: The abbreviation ICARE has not been introduced. Also a reference should be added.

P19, L463: Which figure? Figure 9?

P19, L463: Abbreviation HYSPLIT has not been introduced.

P20, Figure 9: The point and lines are difficult to see. Please try to improve the figure (by e.g. using different colors than the ones currently used).

P21, L473-474: I don't see any blue dots. Do you mean green crosses?

P21, L481-485: This four line paragraph should be combined with the previous or following one.

P21, L482: Add reference to the respective figure.

P21, L500: Which figure? Figure 9 or Figure 8?

P22, L505-514: Combine paragraphs.

P22-23, Conclusion: Make real paragraphs and not one paragraph each 1 or 2 sentences!

P23, 550: What are the implications of your study? A concluding paragraph stating this is missing.

P23, L585: Add this sentence to the previous paragraph.

P24, Figure 10: This figure should be labelled "Fig. A1".

P24-25: Make real paragraphs!

P25, Figure 11: This figure should be labelled as Figure A2.

P25, Figure 11 caption: Provide the date additional to the name of the flight.

P26, L625-626: empty line between the references is missing.

**Technical corrections:**

P1, L31: Hydroxyl -> hydroxyl

P1, L33: space between comma and year missing

P2, L34: Add "e. g." before the references.

P3, L87: Section 2 -> Sect. 2

P3, L97: A last objective -> Another objective

P4, L101: lower the troposphere -> the lower troposphere

P4, L103: add comma before "respectively"

P4, L103-104: replace "in" by "for" and add "campaign" so that it reads "A third campaign is planned for 202672027."

P4, L113: Figure 1 -> Fig. 1

P6, L150-151: Avoid separation of unit and number at the line break.

P6, L133: space between number and unit missing.

P7, L163: add "a" -> a microwave limb sounding system

P7, L176 and 178: add "the" before naming the institutions.

P7, L177: including -> rather "consists of"

P7, L178: add a comma and "respectively" after "2 km".

P10, L232: Change sentence as follows: Once the balloon has passed the convective system, it returns to the initial altitude…..

P16, L335: influences -> influence

P17, L400: Figure -> Fig.

P18, L421: delete "of"

P19, L462: flight -> balloon

P20, L466: flew -> was transported

P20, L49-470: Avoid separation between number and unit.

P20, Figure 9 caption: The dots are rather crosses.

P20, Figure 9 caption: HIMAWARI or Himawari? Use a consistent way of writing.

P21, L467: Dots -> crosses?

P21, L482: add "diagram" after "Hovmöller"

P21, L487: Himawari or HIMAWARI? See my comment on P20, Figure 9 caption.

P21, L488: line break obsolete.

P21, L499: Himawari or HIMAWARI? See my previous comments on this. Use a consistent way of writing.

P22, L514: necessaries -> necessary

---

## Author Response (AR2)

**Editor comments on egusphere-2024-3249**

*Influence of atmospheric waves and deep convection on water vapour in the equatorial lower stratosphere seen from long-duration balloon measurements by Carbone et al.*

Although this manuscript went already through a round of revisions, there are many things left that need to be corrected before the manuscript can be accepted for publication. Thus, another round of major revisions is necessary.

**General comments:**
1. Please consider the ACP manuscript preparation guidelines and prepare your manuscript accordingly.
https://www.atmospheric-chemistry-and
physics.net/submission.html#manuscriptcomposition

In the abstract the introductory part and the concluding part is missing, see the specific guidelines for title, abstract and conclusions:
https://www.atmospheric-chemistry-and-physics.net/policies/guidelines_for_authors.html
Please revise the abstract accordingly. See also my comments given below under specific comments.

Authors : We have rewrite most of the abstract, beginning with a more introductive context, and ending with more conclusive sentences.

2. In the text Section and Figure abbreviated as Sect. and Fig. unless they appear at the begin of the sentence. Please correct this throughout the manuscript.

Authors : Done.

3. Paragraphs should have a reasonable length. Please check all paragraphs and combine the ones
which consists only of 1-3 sentences. The text becomes hardly readable to to this from page 10 on there this has excessively been done,

Done, we have merged paragraphs as much as possible.

4. The whole manuscript feels a bit mixed up. Text that belongs to the results is added in the methodology section and text that belongs to methodology is found in the results. Also the data is described in the wrong order. MLS data is used already before the data and instrument are described. Please sort your text and prepare the manuscript according to the manuscript preparation guidelines.

Authors: We have revised the structure of the manuscript so the reading becomes smoother. Therefore, some of the comments below do not apply anymore.

Specific comments:
P1, L15: An introduction to the field is missing. What is the purpose of this study?

Authors : It was added in the third sentence of the abstract.

P1, L22 and L26: Using the names you have given your flights in the abstract is meaningless for the reader. Use the dates instead.

Authors : We erased the flight names and instead, referred them as "first flight" and "last flight".

P1, L27: The concluding part in the abstract is missing. What are the implications of your study and
results?

A last sentence was added to insist on the significant role played by waves in the water vapour modulation

P2, L60: Has the abbreviation "CPT" been introduced?

Authors : You are right. Cold Point tropopause appears for the first time l49. We've added the acronym there.

P2, L65: change to "that not necessarily reaches the stratosphere, see e.g. Rysman et al., 2017)

Authors : Done.

P4, L105: Don't make paragraphs consisting of one line. Add this sentence to the following paragraph.

Authors : As written before, we have merged several paragraphs belonging to the same subsections.

P4, L111: What do you mean with "the environmental cold temperature"? Do you mean "cold
environmental temperatures"? Please rephrase/correct the sentence.

Authors : To avoid any confusion, we have rephrased "to protect electronics from the lower stratosphere cold temperatures (e.g. Thermodynamic Sensor TSEN)".

P4, L112: What is "TSEN"? The abbreviation has not been introduced.

Authors : Thermodynamic SENsor. It was defined later in Section 3. Now it is defined at its first appearance.

P4, L117ff: Using the flight names here alone is not helpful for the reader. Please add the dates and also add a table summarizing the names and dates for the flights used in your study.

Authors: Done. We added Table 1 listing the flights characteristics.

P4, L122: Who is meant with "They"? The balloons? Please be more precise.

Authors: Yes, we refer to the flights.

P5, L124: Who or what is meant with "it"?

Authors : In the following of the previous sentence, it refers to the figure.

P5, L126: To which figure are you referring to?

Authors: Figure 2, panels b and c.

P5, L127: What do you mean with "the balloons evolved"? Please rephrase,
Authors: We rephrased such as : "All the balloons were flown during…".

P5, Figure 2: MLS used here, but neither the data or instrument have been described yet. Doesn't
this figure rather belong to the result section?
Authors: No, this figure aims to give the context of the flight

P6, L132: What is "SLDA"? The abbreviation has not been introduced.
The SDLA acronym has been added L124.
P6, L138: "regions of strong absorptions within fundamental bands" not clear. Make an own sentence clearly writing what you mean.
Authors: This is about spectroscopy. Fundamental bands being regions of stronger absorption by molecules. We rephrased "These spectral regions depict strong absorptions within fundamental bands."

P6, Section 2.2. (and also previous section): A large part of what is written here goes beyond the pure data description and rather belongs to the methodology section. Move these parts to the methodology section and rename this one to e.g. "Balloon flights and methodology".

P6, L193: It is not clear what kind of methodology you have developed. Is this really new? I have the feeling already known methods are used here.
Authors: we do not claim using a methodology based on anomalies of water vapor is strictly new. However, applying this kind of approach to the Stratéole 2 *in situ* measurements, allows to bring to light the impact of wave and sometimes overshooting deep convection on lower stratospheric water vapour from long duration balloons, which is unprecedented.

P6, L197: How do you know that these are caused by overshooting convection and atmospheric waves? Provide an explanation for the impact of overshooting convection and atmospheric waves on the anomalies and also provide some references.
Authors: Do not apply anymore following the revisions. The meaning here was that the selected temporal extent for the averaging of MLS dataset smooth out the impact of large-scale/medium scale equatorial atmospheric waves (±10 days around each of the balloon nights), which is longer than the longest wave periods as seen from ECMWF. Due to the coarse horizontal resolution of MLS, direct injections from overshooting deep convection are surely not resolved in the vertical profiles of MLS.
Subtracting such a smoothed average of MLS from the *in situ* data removes the tape recorder variability, keeping the information on atmospheric waves' and overshooting convection's modulation in the *in situ* data.

P7, L161: Description of MLS appears after it the data has been used. See my comment above. The sections should be revised and appear in a different order.
Authors: This has been arranged.

P7, L164-165: The sentence "The vertical grid ….." is not clear and should be revised.

Authors: This is how it is explained in the MLS Quality product document.

P7, L171-172: Remove line break. This sentence should be added to the previous paragraph.
Authors: Done.

P7, L182: "……. and by using cloud-type data deduced from cloud-type and phase product" not clear and needs to be revised.
Authors : We revised such as "This algorithm generates CTH products combining data and models. It uses different AHI observations, a radiative transfer model (RTTOV), temperature and humidity vertical profiles (from Numerical Weather Prediction models (NWP)) and cloud-type data deduced from cloud-type and phase product (Kouki et al., 2016)."

P7, L188: Why do you want to evaluate ERA5? Aren't you just comparing these data sets to see if these can be applied together?
Authors: No, we don't compare the *in situ* water vapour anomalies with temperature anomalies from ERA5. The purpose of this section is to show that the ERA 5 temperature products can be used to highlight perturbations due to large-scale atmospheric waves. To be confident, we first demonstrate that the ERA 5 temperatures compare well with measured *in situ* temperatures if we select ERA 5 following colocalization criteria. Then, being confident, we can use temperature anomalies for our analysis. Though we could remove this section, we believe that it gives a bit more confidence in the approach and we prefer to keep it.

P7, L188: Which products are used? List these here explicitly.
Authors: This section has been revised and the sentence completed : "The methodology we have developed relies on the calculation of local anomalies which are obtained as the difference between nighttime i*n situ* water vapour measurements and unbiased MLS v5 water vapour values averaged in the same area, around the same date."

P8, L199: Combine the sentences and replace "indeed" by "since" or "because".
Authors: We prefer to remove "Indeed" since the other proposition do not fit with our meaning.

P8, L201: trapped in what? Clouds? Not clear. Please rephrase sentence,
Authors: This issue of outgassing is well-known in the community of stratospheric water vapour. During the ascent of the balloon through the wet tropical troposphere, and in some cases where the balloon flies through a mixed phase cloud, water vapour but also supercooled water or crystals stick onto the balloon or gondola surfaces. In subsaturated environment, such as the TTL and the stratosphere, the crystals or supercooled dropplets sublimate or evaporate, releasing additional water vapour. In the case of water vapour trapped on the surface during the ascent, the pressure decrease during the ascent favors the outgassing through breaking of weak Van der Walls bonds. We believe that it is not necessary to provide as much details here: it will add additional information which is not useful for the focus of this study.
We therefore propose the following revision: "Daytime measurements are contaminated by outgassing of the (tropospheric) water vapour molecules, or supercooled droplets found while crossing high altitude clouds, which sticks to the balloon or the Zephyr surfaces during

the ascent of the balloon. As the surrounding pressure decreases, the trapped water is released in the environment, especially during daytime."

P8, L205: delete "first" and replace "followed" by "based on".
Authors: Do not apply anymore.

P8, L205: add the criteria here (again) or refer to the previous paragraph/section where these are stated.
Authors: Done.

P9, L216: Introducing ERA5 and ECMWF comes here long after ERA5 has already been mentioned and used. This introduction should be made at the begin of Sect. 3.
Authors: Do not apply anymore.

P9, Figure 4: The green dots and red line are not that good visible. The bullets for the dots should be larger and the red line plotted somewhat thicker.
Authors : We have thickened the red line and grown the dots and added a black border for all dots

P10, L240: Why evaluation? This is rather just a comparison between the data sets.
Authors : The section has been renamed. See our comment above about this section.

P10, L241-249: This text part belongs to Sect. 2.2 where the balloon instrument should be described.
Authors: Done.

Further make paragraphs with a reasonable length and not one paragraph for just one or two sentences.
P11, L266: Which data is shown here? MLS? Please clearly state which data is shown.
Authors: The *in situ* water vapour anomalies are from Pico-STRAT Bi Gaz, since MLS data are remote sensing. We rephrased such as "Longitude/time "quasi-Lagragian" Hovmöller diagrams in temperature anomalies for each flight with their night-time trajectory color-coded as a function of balloon-borne water vapour anomalies from Pico-STRAT Bi Gaz."

P11, Figure 5 caption: You mean the respective balloon trajectory with the MLS water vapour color coded?
Authors: No. We calculated *in situ* water vapour anomalies: from Pico-STRAT Bi Gaz.
In addition, we make it clear in the main text also (line 292-295): "In this section, we explore the impact of atmospheric waves on the modulation of water vapour by studying the correlation between *in situ* water vapour anomalies (from Pico-STRAT Bi Gaz measurements) and ERA5 temperature anomalies." And in the caption of Figure 5 : "…color-coded as a function of balloon-borne water vapour anomalies from Pico-STRAT Bi Gaz."

P11, Figure 5: Which data sets have been used to derive the Hovmöller diagram is not clear. Maybe it would also be good to mention somewhere what a Hovmöller diagram is and for what it is used.

Authors: In the main text (now lines 293-294), we express that the ERA5 3D temperature fields are used to build the Hovmöller diagrams. The details on how the ERA5 temperature anomalies are calculated are given a bit later in the text (now lines 297 -301).

The caption of figure 5 is then revised: "Figure 5: Longitude/time "quasi Lagragian" Hovmöller diagrams in temperature anomalies, calculated from ERA5 3D temperature fields for each flight of Pico-STRAT Bi Gaz. The balloons night-time trajectories are color-coded as a function of balloon-borne water vapour anomalies from Pico-STRAT Bi Gaz. The temperature anomalies are calculated hourly as the difference between the ERA5 temperatures averaged over ±5° around the mean latitude of the balloon for each night and the zonal mean temperature over the same latitude band. (a) C0_05_TTL2. (b) C1_12_STR4 flight. (c) C1_03_TTL4 flight, (d) C1_07_TTL4 flight, (e) C1_15_TTL4 flight."

In addition, we moved the paragraph "Figure 5 shows longitude/time quasi-Lagrangian Hovmöller diagrams of ERA 5 temperature anomalies for the 5 flights of Strateole 2 carrying the Pico-STRAT Bi Gaz instrument. We superimposed *in situ* water vapour anomalies from Pico-STRAT Bi Gaz, color-coded in anomaly amplitude along the balloon trajectory" at a later position in the text.

P12, L268-270: This sentence belongs to the data description section. Rephrase the sentence and refer to the respective section and only point here out the limited resolution of the ERA5 data.

Authors: Done. We also added a new subsection "3.4. ERA 5 temperature fields" (Lines 180-190) which describes ERA 5 products and includes the paragraph L268-270.

P12, L274ff: The entire remainder of the page belongs to the methodology section. Also revise here the paragraphs so that each has a reasonable length.

Authors: We thought it would be more straightforward and understandable to leave this part along in the Wave influence section. This is not the methodology to calculate the temperature anomalies from ERA5 but to highlight the presence of large scale wave along the trajectory of the balloons.

We prefer to keep this part here.

P13, L306: add the date.

Authors: we revised this section so this comment does not apply anymore.

P14, L314: What is "VGWV"? The introduction has not been introduced.

Authors: the definition of this acronym is given Line 332, right before the first mention of this acronym "The sign of this correlation depends on the local vertical gradient of water vapour (hereafter VGWV) at the balloon flight level:…"

P15, L327: Sentence should be added to the previous paragraph.

Authors: Done.

P15, Table 2: This table appears in the middle of the text (last part of the paragraph appears between table and table caption).

Authors: this has been corrected.

P16, L358-361: These sentences should be added to the previous paragraph.
Authors: Done.

P16, L381-382: Add this paragraph either to the previous or following paragraph.
Authors: Done.

P17, L394-395: Again one paragraph for just one sentence.
Authors: Solved: the sentence moved to the next paragraph.

P17, L396: Which data? MLS? The balloon data?
Authors: The anomalies are balloon-borne *in situ* anomalies. This is now specified in the text.

P17, Figure 8 caption: Add that the dates for the flights are given in the figure legend.
Authors: Done.

P18, L407-408: Again one paragraph for just one sentence. This sentence also rather belongs to the figure caption than in the main text.
Authors: Thank you for pointing this out: we removed it.

P18, L412: Do you mean colored in the figure? Be more clear.
Authors: You are right. In fact, this sentence applies to any cases. Then, we revised this sentence such that: "A linear behaviour of the balloon-borne water vapour anomalies with potential temperature would indicate that water vapour variations are mainly dominated by the displacement of the balloons in the vertical gradient of water vapour even in the vicinity of deep convection."

P18, L412: Sentence "displacement of the balloons in the vertical gradient of water vapour " not clear. What do you mean? Another flight showing a different case? Please rephrase.
Authors: We have rephrased this sentence. It means exactly what was described in section 4.1 about vertical displacement though it was badly written.
We revised such as: "A linear behaviour of the balloon-borne water vapour anomalies with potential temperature would indicate that water vapour variations are mainly dominated by the vertical displacement of the balloons. This is due to the vertical gradient of water vapour, such as detailed in the section 4.1 about atmospheric waves influence."

P18, L422-425: Where is this shown? To which figure are you referring to?
Authors: We can see the cold perturbation of low-amplitude large-scale wave on Fig. 5d (the reference to this figure is now added in the text). We also revised this paragraph such as: "The only exception for the dry cases is the night of November 13, 2021, of C1_07_TTL4:  the Hovmöller diagram for this night indicates a very weak wave signature (very weak cold temperature anomalies, see Fig. 5d), for which the calculated vertical displacement is too small to explain amplitude of the dry anomalies. The additional vertical displacement of isentropes due to convection which is overpassed by the balloon could be an explanation since upward vertical motion would bring lower mixing ratios from below. Another possible explanation would be the impact of a small wavelength gravity wave generated by the overpassed deep convective system that is not resolved in ERA5, nor seen in the Hovmöller

diagram. The perturbation produced by the gravity wave can cumulate on the top of the cold perturbation resolved in the Hovmöller diagram."

P18, L426: Sentence "Another case of flight C1_15_TTL4 should be underlined" not clear and should be rephrased.
Authors: this has been revised such as : "Similarly, the case of December 5, 2021, of flight C1_15_TTL4 depicts dry balloon-borne anomalies, originating from the cumulative effect of atmospheric waves of different scale." (L447).

P19, L435: What is HIMAWARI? The abbreviation has not been introduced.
Authors : Himawari is not an abbreviation, it is the name of a satellite meaning "Sunflower" in Japanese. We do not think useful to specify this. Additionally, the Himawari cloud top products which are used in the study are described in a specific section 3.3 (page 6).

P19, L441: The abbreviation CALIOP has not been introduced. Also it should be mentioned what kind of instruments these are.
Authors : You are right. CALIOP (with its acronym) is now introduced in a short new subsection 3.5. So the acronym is not specified anymore p19.

P19, L457: Add some more details on the storm Rai (when did it form, how long did it last, from where to where did it move and what implications had this storm).
Authors: Details on the storm Rai are already given in the beginning of section 4.2. (L400).

P19, L460: The abbreviation ICARE has not been introduced. Also a reference should be added.
Authors: ICARE is one of the 4 poles of the AERIS French e-infrastructure. The meaning of ICARE has been found to be "Cloud-Aerosol-Water-Radiation Interactions".
We revised the sentence such as "The time and date of the cloud top image is chosen to be the closest to the time when the water vapour anomaly is the highest. Cloud top products are made available on the French AERIS/ICARE (Cloud-Aerosol-Water-Radiation Interactions) datacenter."

P19, L463: Which figure? Figure 9?
Authors: Yes, figure 9a. This is corrected.

P19, L463: Abbreviation HYSPLIT has not been introduced.
Authors: Done (L485).

P20, Figure 9: The point and lines are difficult to see. Please try to improve the figure (by e.g. using different colors than the ones currently used).

Figure 9 has been improved using blue color scale for the cloud tops.

P21, L473-474: I don't see any blue dots. Do you mean green crosses?
Authors: No, in fact it was referring to Fig 8 again, panel c. The paragraph has been revised such as:

"It is worth noticing that the signature of December 13, 2021, in Fig. 8c (pink dots) may also be under the influence of Raï, with anomalies above 0.5 ppmv and out of the main scatterplot at 414 K. The balloon position close to Raï's eye on December 12, 2021, induced a depressurization. Though on the edge of the scatter plot, the water anomaly distribution on Fig. 8c (blue dots) is linear and not typical of an overshoot signature."

P21, L481-485: This four line paragraph should be combined with the previous or following one.
Authors: We included the sentence to the following paragraph.

P21, L482: Add reference to the respective figure.
Authors: Done, it was Fig. 5b.

P21, L500: Which figure? Figure 9 or Figure 8?
Authors: This is Fig. 9d. We corrected it.

P22, L505-514: Combine paragraphs.
Authors: Done.

P22-23, Conclusion: Make real paragraphs and not one paragraph each 1 or 2 sentences!
P23, 550: What are the implications of your study? A concluding paragraph stating this is missing.
Authors: We added a sentence at the end of the wave paragraph to insist on the role of waves in the H2Ovap modulation L563-565."These statistics show that atmospheric waves are an important process, or even the main process for some flights, driving the modulation of water vapor during Stratéole 2". Concerning the convection part, we let the conclusion as it is, this we think it summarizes the main results of our study. We end we a more general sentence about our study "More generally, this study uses long duration balloon measurements and an approach based on *in situ* anomalies of water vapour. We demonstrate that they are useful tools to study the impact of large scale waves as well as very intense deep convection on the lower stratospheric water vapour abundance all around the equatorial belt."

"

P23, L585: Add this sentence to the previous paragraph.
Authors: Done.

P24, Figure 10: This figure should be labelled "Fig. A1".
Authors: Done.

P24-25: Make real paragraphs!
Authors: We reorganized the paragraph properly.

P25, Figure 11: This figure should be labelled as Figure A2.
Authors: Done.

P25, Figure 11 caption: Provide the date additional to the name of the flight.
Authors : We have changed night number to the dates.

P26, L625-626: empty line between the references is missing.
Authors : References have been updated.

Technical corrections:
P1, L31: Hydroxyl -> hydroxyl
Done
P1, L33: space between comma and year missing
Done
P2, L34: Add "e. g." before the references.
Done
P3, L87: Section 2 -> Sect. 2
Done
P3, L97: A last objective -> Another objective
P4, L101: lower the troposphere -> the lower troposphere
done
P4, L103: add comma before "respectively"
done
P4, L103-104: replace "in" by "for" and add "campaign" so that it reads "A third campaign is planned for 2026/2027."
Done
P4, L113: Figure 1 -> Fig. 1
done
P6, L150-151: Avoid separation of unit and number at the line break.
ok
P6, L133: space between number and unit missing.

P7, L163: add "a" -> a microwave limb sounding system
done
P7, L176 and 178: add "the" before naming the institutions.
Done
P7, L177: including -> rather "consists of"
Done
P7, L178: add a comma and "respectively" after "2 km".
Done
P10, L232: Change sentence as follows: Once the balloon has passed the convective system, it returns to the initial altitude…..
Done

P16, L335: influences -> influence
Done
P17, L400: Figure -> Fig.
P18, L421: delete "of"
P19, L462: flight -> balloon
P20, L466: flew -> was transported

Done
P20, L469-470: Avoid separation between number and unit.

P20, Figure 9 caption: The dots are rather crosses.

Now, these are dots.

P20, Figure 9 caption: HIMAWARI or Himawari? Use a consistent way of writing.

Done, Himawari is now chosen everywhere.

P21, L467: Dots -> crosses?

No. We are referring to Fig. 8 where nights of depressurization are indicated with colored dots.

P21, L482: add "diagram" after "Hovmöller"

P21, L487: Himawari or HIMAWARI? See my comment on P20, Figure 9 caption.

P21, L488: line break obsolete.

P21, L499: Himawari or HIMAWARI? See my previous comments on this. Use a consistent way of

writing.

P22, L514: necessaries -> necessary

---

## Editor Decision (ED2)

P1, L20: flown under not correct, I guess you mean flown "on". Aren't this instruments mounted on the balloon?

P1, L24-25: "for a flight which flew" -> this sentence makes no sense. Rephrase it to "for a flight performed" or "for a flight made".

P1, L26: What do you mean with "variabilities according to the flight"?

P1, L20-27: The abstract still needs to be improved, especially the result part and implications of this study. The last sentence of the abstract is really a bit poor. It sounds like that waves are only important for the measurements on balloons although the observations of these waves are important to understand atmospheric processes related to water vapour.

P3, L86: Acronym MLS has not been introduced.

P3, L98: two first -> first two

P4, L108: Add "the" -> the stratospheric water vapour budget

P4, L109: What do you mean with "five flights have been already released"? Shouldn't it rather read "five flights have already been performed" although I think that "already" is generally obsolete in the sentence.

P5, L122: Delete "used" and consider even renaming the section to just " Instruments and Datasets".

P5, L124 and following subsections: 3.1. -> 3.1, the dot after the second number is obsolete

P5, L142: space between number and unit missing.

P8, L215 and throughout the manuscript: In many occasions there is a double space between "correlation" and "coefficient" -> one space should be removed.

P8, L215: to water vapour -> of water vapour

P10, L262-263: from 800 m to 1.5 km  is an increase and not a drop of the balloon altitude.

P10, L270: wave -> waves

P11, L274: Remove full stop at the end of the subsection header.

P12, Figure 5 caption: "for each flight" misplaced? Doesn't this hold for both values and thus this should appear at the end of the sentence?

P14, Figure 6 caption, last line: Shouldn't "respectively" rather appear at the end of the sentence? However, I would rather suggest to omit using "respectively".

P15, L333: Here 0.65 written, but in the table (Table 2) 0.64 listed.

P15, L339: fight -> flights

P15, L340: In this sentence something is missing. What is the case for the other two flights.

P15, L350: stands in -> is in

P15, L350: observed in the -> obtained from

P15-16: Following your discussion on the results is really difficult. More guidance to which table you are referring to would maybe be helpful and generally improving the structure and better motivate for what you actually do which analysis.

P16, L380: To which table are you referring here to? What about Table 3? This table is only shortly mentioned, but not further discussed.

P17, L397: Rai is one -> Rai was one

P18, L417: If you cannot derive the anomalies, which anomalies are then plotted in Fig. 8? How have these been derived?

P18, Figure 8: Hard to see here a quasi-linear relationship. What may help is if you would adjust the y-axis of panel and b to the y-axis of panel c. Since the black dots overlay over all other coloured dots it is quite hard to follow your discussion later on. It would help if you would use a different colour than black for the data of this date. Further, what date is plotted with the black dots is missing in the legend of panel a and c.

P19, L425: Shorten section title? Thus, skip "Isentropic level"?

P19, L432: Add "the" before "balloon".

P19, L438: Add "the" before "amplitude".

P20, L485: was fed -> was run or was driven

P21, L490: What is meant with signature? Do you mean the structure in the anomalies? Do you mean "positive anomalies"?

P22, L493: The blue dots are hardly visible in the figure.

P22, L497: Strateole is written in at least three different ways throughout the manuscript. Decide for one way of writing and do this consistently throughout the manuscript.

P22, L508: are limit cases to the -> are limiting cases of the

P23, L533: I suggest to rename "Conclusion" to "Summary and Conclusion" since you also provide a summary of your results.

P23, L535: under -> on

P23, L545: flight -> flights

P24, L569: Change "More generally" to "To summarize" or "To conclude".

P24, L578: bias -> biases

P24, L579: If -> Although

P25, L606: robufstness -> robustness

P26, L607: There is no next section!

P26, L608: Change TTL2 to the full flight name.

P25 and P26, Figure A1 and A2 caption: Blue text should be black text.

---

## Author Response (AR3)

Dear editor,

Please to find our replies to your comments. The manuscript has been revised accordingly.

Best regards,

Sullivan Carbone and co-authors
* * *
P1, L20: flown under not correct, I guess you mean flown "on". Aren't this instruments mounted on the balloon?

Authors: No, the instruments are flown hanging down below the balloon. "under" is commonly used in this case.

P1, L24-25: "for a flight which flew" -> this sentence makes no sense. Rephrase it to "for a flight performed" or "for a flight made".

Authors: The sentence has been rephrased such as "This is the case for one of the flights which flew over the Raï typhoon".

P1, L26: What do you mean with "variabilities according to the flight"?

Authors: We mean that the rate of nights influenced by atmospheric waves varies depending on the flight. We rephrased such as "with variabilities in this rate according to the flight".

P1, L20-27: The abstract still needs to be improved, especially the result part and implications of this study. The last sentence of the abstract is really a bit poor. It sounds like that waves are only important for the measurements on balloons although the observations of these waves are important to understand atmospheric processes related to water vapour.

Authors: It has been changed as follows: Most of atmospheric species enter the stratosphere through the tropical tropopause layer, a place of interplay between many processes of different scales. Water vapour ($H_2O_{vap}$) is a key compound in this layer and its entry into the tropical stratosphere is crucial for stratospheric chemistry and climate. In this paper, we present a methodology based on the calculation of in situ $H_2O_{vap}$ and temperature anomalies to estimate the modulation of $H_2O_{vap}$ due to atmospheric waves and deep convection. $H_2O_{vap}$ were obtained from *in situ* measurements of five Pico-Strat Bi Gaz spectrometers which were flown under long-duration balloons during the Strateole 2 campaigns. The calculation of Pearson's correlation coefficients is performed between averaged ERA5 reanalysis temperatures and in situ $H_2O_{vap}$ anomalies. In case of a monotonic vertical gradient of $H_2O_{vap}$, the absolute value of the correlation coefficient is high (typically 0.65). For the other flights, we highlight lower correlations, due to changes in time of the vertical gradient of stratospheric $H_2O_{vap}$, and large convective systems overshooting the tropopause. This is the case for one of the flights which flew over the Raï typhoon (correlation coefficient of 0.31 due to both contributions). Depending on the flights, we also show that for 47 % up to 70 % of the probed nights, $H_2O_{vap}$ anomalies can be explained by atmospheric waves, which highlights the major role played by waves on $H_2O_{vap}$ in

the TTL . We also show that long duration balloons measurements are precious to highlight overhsooting signature of $H_2O_{vap}$ in the upper TTL.

P3, L86: Acronym MLS has not been introduced.

Authors: Done.

P3, L98: two first -> first two

Authors: Done.

P4, L108: Add "the" -> the stratospheric water vapour budget

Authors: Done.

P4, L109: What do you mean with "five flights have been already released"? Shouldn't it rather read "five flights have already been performed" although I think that "already" is generally obsolete in the sentence.

Authors: We mean that additional flights of Pico-STRAT Bi Gaz will be carried out in the next (and last) campaign of Strateole 2. We rephrased such as:

"Five Pico-STRAT have been already released during the first two campaigns, allowing to measure *in situ* water vapour, $CO_2$ and $CH_4$ mixing ratios. Additional flights of Pico-STRAT Bi Gaz will take place during the last campaign of Strateole 2 in the end of 2026."

P5, L122: Delete "used" and consider even renaming the section to just " Instruments and Datasets".

Authors: Done.

P5, L124 and following subsections: 3.1. -> 3.1, the dot after the second number is obsolete

Authors: Done.

P5, L142: space between number and unit missing.

Authors: Done.

P8, L215 and throughout the manuscript: In many occasions there is a double space between "correlation" and "coefficient" -> one space should be removed.

Authors : The space is due to the paragraph format as "justified".

P8, L215: to water vapour -> of water vapour

Authors: Done.

P10, L262-263: from 800 m to 1.5 km is an increase and not a drop of the balloon altitude.

Authors: Done.

P10, L270: wave -> waves

Authors: Done.

P11, L274: Remove full stop at the end of the subsection header.

Authors: Done.

P12, Figure 5 caption: "for each flight" misplaced? Doesn't this hold for both values and thus this should appear at the end of the sentence?

Authors: it has been changed as follows: Longitude/time "quasi-Lagragian" Hovmöller diagrams in temperature anomalies, calculated from ERA5 3D temperature fields for each Pico-STRAT Bi Gaz flight.

P14, Figure 6 caption, last line: Shouldn't "respectively" rather appear at the end of the sentence? However, I would rather suggest to omit using "respectively".

Authors: Done.

P15, L333: Here 0.65 written, but in the table (Table 2) 0.64 listed.

Authors: Done.

P15, L339: fight -> flights

Authors: Done.

P15, L340: In this sentence something is missing. What is the case for the other two flights.

Authors: We rephrased such as:

"All the 2021 fights overflew the Maritime Continent and two of them (C1_07_TTL4 and C1_15_TTL4) overflown it during these 8 days."

P15, L350: stands in -> is in

Authors: Done.

P15, L350: observed in the -> obtained from

Authors: Done.

P15-16: Following your discussion on the results is really difficult. More guidance to which table you are referring to would maybe be helpful and generally improving the structure and better motivate for what you actually do which analysis.

Authors: we revised the discussion about the Pearsons'r coefficient to improve clarity.

Lines 343-354: "A negative $r_{H2O\ T}$, such as for the C1_03_TTL4 and C1_12_STR4 flights, does not rule out the signature of atmospheric waves in the water vapour modulation. Since atmospheric waves produce temperature anomalies, another way to verify the effective influence of atmospheric waves is to compute the temperature-temperature Pearson's correlation coefficient ($r_{T\ T}$) between ERA5 temperature anomalies and *in situ* air temperature observed by Pico-STRAT BI Gaz (see Table 3). The $r_{T\ T}$ listed in Table 3 are high for all the flights: the mean Pearson's correlation coefficient is 0.77 ± 0.06. This indicates that even flights which depict near-zero or negative $r_{H2O\ T}$ are strongly influenced by atmospheric waves.

During the period from December 28, 2021, to January 9, 2022, the C1_15_TTL4 balloon flew in a similar structure of the tape recorder as the C0_05_TTL2 balloon (i.e. positive vertical gradient), which has the highest $r_{H2O\ T}$ (0.64). One can thus expect similar Pearson's correlation coefficient for both flights during this period. Indeed, restricting the calculation of the Pearson's correlation coefficient to the period from December 28, 2021, to January 9, 2022, leads to a $r_{H2O\ T}$ of 0.65, very similar to the one obtained for C0_05_TTL2. In both cases, the Pearson's correlation coefficient therefore is in the 0.6-0.7 range, which is the highest value obtained so far for the Strateole 2 flights."

P16, L380: To which table are you referring here to? What about Table 3? This table is only shortly mentioned, but not further discussed.

Authors: The statistics provided here do not refer to any table.

P17, L397: Rai is one -> Rai was one

Authors: Done.

P18, L417: If you cannot derive the anomalies, which anomalies are then plotted in Fig. 8? How have these been derived?

Authors: This figure shows only datapoints obtained out of depressurization events. Depressurizations only occur on a portion of a given night (2-3 hours) and occur whenever the balloon overpass a deep convective system. Whenever a balloon experience depressurization during one given night, it means the measurements during this given night (before or after the depressurization) can potentially be influenced by the deep convective system beneath. Those nights are highlighted in color on Figure 8. One exception is found for the night of November 25, 2021 (Fig. 8b) for which no depressurization occurred.

To clarify, we revised the text such as:

Lines 417-424: "Often, before and after a depressurization, the water vapour anomalies, whether dry or wet, are large. We thus use here depressurization cases as a proxy of deep convection flyover. Whenever a balloon experience depressurization during one given night, the water vapour measurements during this given night can potentially be influenced by the deep convective system beneath. Figure 8 shows scatter plots of balloon-borne *in situ* water vapour anomalies as a function of the potential temperature for the three flights that have experienced depressurizations. Please note that the measurements during the balloon depressurizations are not used to compute water vapour anomalies because it would not be possible to compute anomalies associated with a fast variation of altitude. Only datapoints obtained out of the depressurization for a given day are plotted."

P18, Figure 8: Hard to see here a quasi-linear relationship. What may help is if you would adjust the y-axis of panel and b to the y-axis of panel c. Since the black dots overlay over all other coloured dots it is quite hard to follow your discussion later on. It would help if you would use a different colour than black for the data of this date. Further, what date is plotted with the black dots is missing in the legend of panel a and c.

Authors: We have changed the black dots to grey dots, we have enlarged the color dots adding black contours and we have added the linear trends (cyan straight line) of some dot clouds .

P19, L425: Shorten section title? Thus, skip "Isentropic level"?

Authors: the influence of deep convection is here divided into 2 mechanisms: the vertical displacement of isentropes and direct injections. Therefore, the subsection titles are then revised to be "5.2.1 Vertical displacement of isentropes" and "5.2.2 Direct injection".

P19, L432: Add "the" before "balloon".

Authors: Done.

P19, L438: Add "the" before "amplitude".

Authors: Done.

P20, L485: was fed -> was run or was driven

Authors: We selected "run".

P21, L490: What is meant with signature? Do you mean the structure in the anomalies? Do you mean "positive anomalies"?

Authors: Here we mean the local water vapour enhancement. We corrected this in the text (L497).

P22, L493: The blue dots are hardly visible in the figure.

Authors: We have enlarged the color dots and we have added black contours.

P22, L497: Strateole is written in at least three different ways throughout the manuscript. Decide for one way of writing and do this consistently throughout the manuscript.

Authors: This has been corrected.

P22, L508: are limit cases to the -> are limiting cases of the

Authors: Done.

P23, L533: I suggest to rename "Conclusion" to "Summary and Conclusion" since you also provide a summary of your results.

Authors: Done.

P23, L535: under -> on

Authors: Pico-STRAT Bi Gaz is located under the balloon on the flight chain. Balloon scientists often use "under" for this case. We kept it like that.

P23, L545: flight -> flights

Authors: Done.

P24, L569: Change "More generally" to "To summarize" or "To conclude".

Authors: Done.

P24, L578: bias -> biases

Authors: Done.

P24, L579: If -> Although

Authors: Done.

P25, L606: robufstness -> robustness

Authors: Done.

P26, L607: There is no next section!

Authors: it has been removed.

P26, L608: Change TTL2 to the full flight name.

Authors: Done.

P25 and P26, Figure A1 and A2 caption: Blue text should be black text.

Authors: Done.